# The murine catecholamine methyltransferase mTOMT is essential for mechanotransduction by cochlear hair cells

Christopher L Cunningham[1], Zizhen Wu[1], Aria Jafari[2], Bo Zhao[3], Kat Schrode[4], Sarah Harkins-Perry[5], Amanda Lauer[4], Ulrich Müller[1]*

[1]The Solomon Snyder Department of Neuroscience, Johns Hopkins University, Baltimore, United States; [2]Department of Surgery, University of California, San Diego, San Diego, United States; [3]Department of Otolaryngology Head and Neck Surgery, Indiana University School of Medicine, Indianapolis, United States; [4]Department of Otolaryngology, Johns Hopkins University, Baltimore, United States; [5]Department of Molecular and Cellular Biology, The Scripps Research Institute, La Jolla, United States

**Abstract** Hair cells of the cochlea are mechanosensors for the perception of sound. Mutations in the *LRTOMT* gene, which encodes a protein with homology to the catecholamine methyltransferase COMT that is linked to schizophrenia, cause deafness. Here, we show that *Tomt/Comt2*, the murine ortholog of *LRTOMT*, has an unexpected function in the regulation of mechanotransduction by hair cells. The role of mTOMT in hair cells is independent of mTOMT methyltransferase function and mCOMT cannot substitute for mTOMT function. Instead, mTOMT binds to putative components of the mechanotransduction channel in hair cells and is essential for the transport of some of these components into the mechanically sensitive stereocilia of hair cells. Our studies thus suggest functional diversification between mCOMT and mTOMT, where mTOMT is critical for the assembly of the mechanotransduction machinery of hair cells. Defects in this process are likely mechanistically linked to deafness caused by mutations in *LRTOMT/Tomt*.

*For correspondence: umuelle3@ jhmi.edu

## Introduction

Hair cells in the mammalian cochlea convert sound-induced vibrations into mechanical signals to enable our sense of hearing. Each hair cell contains at the apical surface a bundle of stereocilia that are organized in rows of decreasing height (*Alagramam et al., 2011*; *Schwander et al., 2010*). Tip links connect the stereocilia of hair cells in the direction of their mechanical sensitivity and are thought to transmit force onto transduction channels that are situated near the lower end of tip links (*Figure 1A*) (*Assad et al., 1991*; *Pickles et al., 1991*). Tip links consist of CDH23 homodimers that interact in trans with PCDH15 homodimers to form the upper and lower part of tip links, respectively (*Figure 1A*) (*Ahmed et al., 2006*; *Kazmierczak et al., 2007*; *Siemens et al., 2004*; *Söllner et al., 2004*). The adaptor proteins harmonin and SANS, and the molecular motor myosin 7a (MYO7A), bind to the CDH23 cytoplasmic domain (*Adato et al., 2005*; *Bahloul et al., 2010*; *Boëda et al., 2002*; *Siemens et al., 2002*) and co-localize near the cytoplasmic domain of CDH23 at the upper tip-link density (UTLD) (*Figure 1A*), an electron-dense area near the insertion site of CDH23 into the stereociliary membrane (*Grati and Kachar, 2011*; *Grillet et al., 2009*). TMHS/LHFPL5, TMIE and TMC1/2 are localized near PCDH15 at the lower tip-link density (LTLD) (*Figure 1A*), an electron

**Figure 1.** Generation of *Tomt* mutant mice. (**A**) Schematic of the tip-link complex in hair cell stereocilia. (**B**) Schematic comparison of mouse TOMT (mTOMT, Uniprot AIY9I9) and mouse COMT (mCOMT, Uniprot O88587). Putative methyltransferase domain is indicated in black. Amino acid numbers are below. Consensus methyltransferase sequence (CMS, [*Vidgren et al., 1994*]) is depicted in yellow. Transmembrane domain (TM) from membrane-bound isoform of COMT is in green. (**C**) Conserved amino acid residues in consensus

*Figure 1 continued on next page*

*Figure 1 continued*
methyltransferase sequence of mTOMT and mCOMT. (D) Genomic locus of mouse *Tomt* (NCBI gene 791260), located on chromosome 7. Four *Tomt* exons are depicted as open boxes, with coding exons in blue. Introns are shown as solid black line. Exon 2, the first coding exon, is enlarged below, with relative locations indicated for ATG, clustered regularly interspaced short palindromic repeats (CRISPR) sgRNA recognition site, and *add* N-ethyl-N-nitrosourea (ENU) mutation site (*Du et al., 2008*). At bottom is *Tomt* exon 2 sequence showing CRISPR sgRNA and protospacer adjacent motif (PAM), and site of Cas9 cleavage (scissors). (E) Schematic of *Tomt* mRNA (NCBI NM_001081679.1) and location of mutations. *Tomt* exons are indicated with green arrows. Consensus *Tomt* coding sequence (CDS, NCBI CCDS40044.1) is in red. Location of *add* mutation is indicated with lightning bolt. Two unique deletions were identified in founder mice after CRISPR/Cas9 pronuclear injection of *Tomt*-specific sgRNA and Cas9. Genetic mouse lines were generated containing either a 12 base-pair (bp) deletion (*Tomt$^{D12}$*) or 77 bp deletion (*Tomt$^{del}$*) in Exon 2 of *Tomt*. Deletions are indicated as dashed-line boxes. (F) Genomic DNA sequencing results from *Tomt$^{+/+}$*, *Tomt$^{add/add}$*, and *Tomt$^{D12/D12}$* mice demonstrating mutations. (G) PCR of *Tomt* genomic DNA from *Tomt$^{+/+}$*, *Tomt$^{del/+}$*, and *Tomt$^{del/del}$* demonstrating 77 bp deletion. (H) RT-PCR results for *Tomt* and *Gapdh* from inner ear tissue from *Tomt$^{del/+}$* (+/−) and *Tomt$^{del/del}$* (−/−). (I) Predicted protein structure of wild-type and mutant TOMT. From top: wild-type TOMT; *add* mutation (R48L); CRISPR 12 bp in-frame deletion (*Tomt$^{D12}$*, H22_R25 del) leading to loss of four amino acids; and CRISPR 77 bp deletion (*Tomt$^{del}$*, R25Qfs*20) leading to a frame-shifted amino acid sequence, premature stop codon and truncated protein.

dense area at the tips of stereocilia (*Kurima et al., 2015*; *Xiong et al., 2012*; *Zhao et al., 2014*). TMHS/LHFPL5, TMIE and TMC1/2 are candidate proteins to be components of the mechanotransduction channel in hair cells (*Kawashima et al., 2011*; *Kim et al., 2013*; *Pan et al., 2013*; *Xiong et al., 2012*; *Zhao et al., 2014*). Thus, the mechanotransduction machinery of hair cells shows a striking asymmetry with distinct protein complexes present at the upper and lower end of tip links.

We know little about the transport and targeting mechanisms that regulate the precise configuration of proteins within the tip-link complex. Stereocilia contain bundles of parallel actin filaments with their barbed ends facing toward the tips of stereocilia. No vesicles have been observed within stereocilia. Membrane proteins and cytoplasmic components are thus thought to be transported into stereocilia at least in part by actin-based molecular motors of the myosin family (*Belyantseva et al., 2005*; *Senften et al., 2006*). Accordingly, MYO7A is required for the localization of harmonin, SANS and PCDH15 within stereocilia (*Bahloul et al., 2010*; *Boëda et al., 2002*; *Senften et al., 2006*). MYO1C binds to CDH23 and is a candidate to participate in CDH23 transport (*Siemens et al., 2004*). The extent to which myosin motor proteins participate in the transport of TMHS/LHFPL5, TMIE, and TMC1/2 is not known, but recent studies have shown that the tetraspan protein TMHS/LHFPL5facilitates the transport of both PCDH15 and TMC1 into stereocilia (*Beurg et al., 2015*; *Xiong et al., 2012*). However, we have only a very limited understanding of the mechanisms by which different proteins control the transport and retention of proteins within the tip-link complex.

Recent studies have shown that mutations in the human *LRTOMT* gene are associated with profound non-syndromic hearing loss at the DFNB63 locus (*Ahmed et al., 2008*; *Du et al., 2008*). *LRTOMT* appears to have evolved from the fusion of two neighboring ancestral genes *LRRC51* and *TOMT*. The former gene has no known function, while the latter is a homolog of COMT, which encodes a catechol-O-methyltransferase implicated in the etiology of schizophrenia (*Männistö and Kaakkola, 1999*; *Tunbridge, 2010*). COMT regulates levels of catecholamines such as epinephrine, norepinephrine, and dopamine in the brain and periphery (*Männistö and Kaakkola, 1999*). *LRTOMT* has two alternative reading frames that encode two different proteins named LRTOMT1 and LRTOMT2. Only the latter isoform encodes a protein with predicted enzymatic activity (*Ahmed et al., 2008*). *Lrrc51* and *Tomt/Comt2* exist in rodents as alternative genes that are located adjacent on the same chromosome. However, no fusion transcripts have been observed between the two murine genes ([*Ahmed et al., 2008*] and our unpublished observations). In the following, we will refer to *Tomt/Comt2* with its official gene name *Tomt*. Recombinant mouse TOMT(mTOMT) has catechol-O-methyltransferase activity and a point mutation in the murine gene that reduces this activity causes deafness (*Du et al., 2008*). Some but not all the mutations in the human *LRTOMT* gene that cause deafness are also predicted to affect methyltransferase activity (*Ahmed et al., 2008*), although this has so far not been demonstrated experimentally. However, the mechanisms by

which mutations in *LRTOMT* and *Tomt* cause deafness are currently unknown and the extent to which catecholamines play a role in this process remains to be established.

Using genetically modified mice generated by ENU mutagenesis and CRISPR-mediated gene editing, we have now investigated the mechanisms by which *Tomt* regulates auditory function. Surprisingly, we demonstrate that *Tomt* is essential for mechanotransduction by hair cells, where it is required for the localization of some components of the mechanotransduction machinery of hair cells to the mechanically sensitive stereocilia. Using mutational analysis, we provide evidence that the function of *Tomt* in mechanotransduction is independent of its enzymatic function. Instead, mTOMT binds to components of the mechanotransduction machinery and our data are consistent with a role for mTOMT in protein transport. Our studies thus suggest functional diversification between mCOMT and mTOMT, where mTOMT has acquired a new role in hair cells that is independent of its methyltransferase activity but critical for the assembly of the mechanotransduction machinery of hair cells.

## Results

### Generation of *Tomt*-deficient mice and measurement of auditory function

mTOMT is a homologue of mCOMT with similarity in the overall domain structure of the two proteins and conservation of key amino acids in the methyltransferase domains (*Figure 1B,C*). mCOMT has two isoforms, one of which contains a transmembrane domain, but the extent to which mTOMT has a transmembrane domain is unclear; some algorithms predict a potential transmembrane domain while others do not (*Figure 1B*; and data not shown).

In order to investigate the function of *Tomt* in the inner ear, we took advantage of *add* mice that we previously generated in an ENU mutagenesis screen (*Du et al., 2008*). *Add* mice carry a point mutation (R48L) in the *Tomt* gene (referred to as *Tomt$^{add}$*) (*Figure 1D,F,I*). Our previous studies have shown that recombinant mTOMT has catechola-O-methyltransferase activity in vitro, although the methyltransferase activity is significantly less compared to mCOMT. The *add* mutation reduces methyltransferase activity but could also affect protein stability (*Du et al., 2008*). To more thoroughly investigate the function of *Tomt*, we wanted to generate a null allele of the *Tomt* gene using CRISPR-mediated gene editing. We therefore designed a guide RNA targeting the first coding exon (exon 2) of the *Tomt* gene (*Figure 1D*) and injected the guide RNA together with Cas9 into fertilized murine zygotes on the C57BL/6 background. Offspring from injections were bred and screened by PCR for genetic modifications at the *Tomt* locus. One of the CRISPR-engineered alleles, referred to as *Tomt$^{D12}$*, carries a 12 base pair (bp) in-frame deletion near the N-terminus (*Figure 1E–I*). The second allele, *Tomt$^{del}$*, carries a deletion of 77 bp leading to a frame-shift and premature stop codon near the N-terminus of mTOMT (*Figure 1E–I*). Analysis of mRNA from *Tomt$^{D12/D12}$* and *Tomt$^{del/del}$* mice confirmed the presence of the mutations in the spliced transcripts (*Figure 1H*, and data not shown). While the *Tomt$^{D12}$* mutation is predicted to produce a protein with an internal deletion of 4 amino acids, the *Tomt$^{del}$* mutation is predicted to generate a functional null allele.

### Analysis of hearing function in *Tomt$^{D12/D12}$* and *Tomt$^{del/del}$* mice

Next, we carried out measurements of the auditory brain stem response (ABR) of homozygous offspring of *Tomt$^{add/add}$*, *Tomt$^{D12/D12}$* and *Tomt$^{del/del}$* mice at 3–5 weeks of age. The ABR thresholds to broadband click stimuli of control wild-type animals were at 26 ± 3 dB (*Figure 2A*). In contrast, ABR responses to click stimuli were absent in all mutant mouse lines even at 90 dB, demonstrating that the mice were profoundly deaf (*Figure 2A*). Recordings of ABRs in response to pure tones revealed that homozygous *Tomt$^{add/add}$*, *Tomt$^{D12/D12}$* and *Tomt$^{del/del}$* mice were deaf across the entire analyzed frequency spectrum (*Figure 2B*). Hearing function was not significantly affected in 4-week-old heterozygous *Tomt$^{add/+}$*, *Tomt$^{D12/+}$* and *Tomt$^{del/+}$* mice (data not shown), consistent with a recessive mode of inheritance of the auditory phenotype.

We next recorded distortion product otoacoustic emissions (DPOAEs), which are mechanical distortions generated in the inner ear when two primary tones ($f_1$ and $f_2$) are presented. Outer hair cells (OHCs) amplify the distortions and they are propagated back through the middle ear and ear canal and can be measured in sound pressure waveforms (*Kemp, 1978*; *Shaffer et al., 2003*). DPOAEs

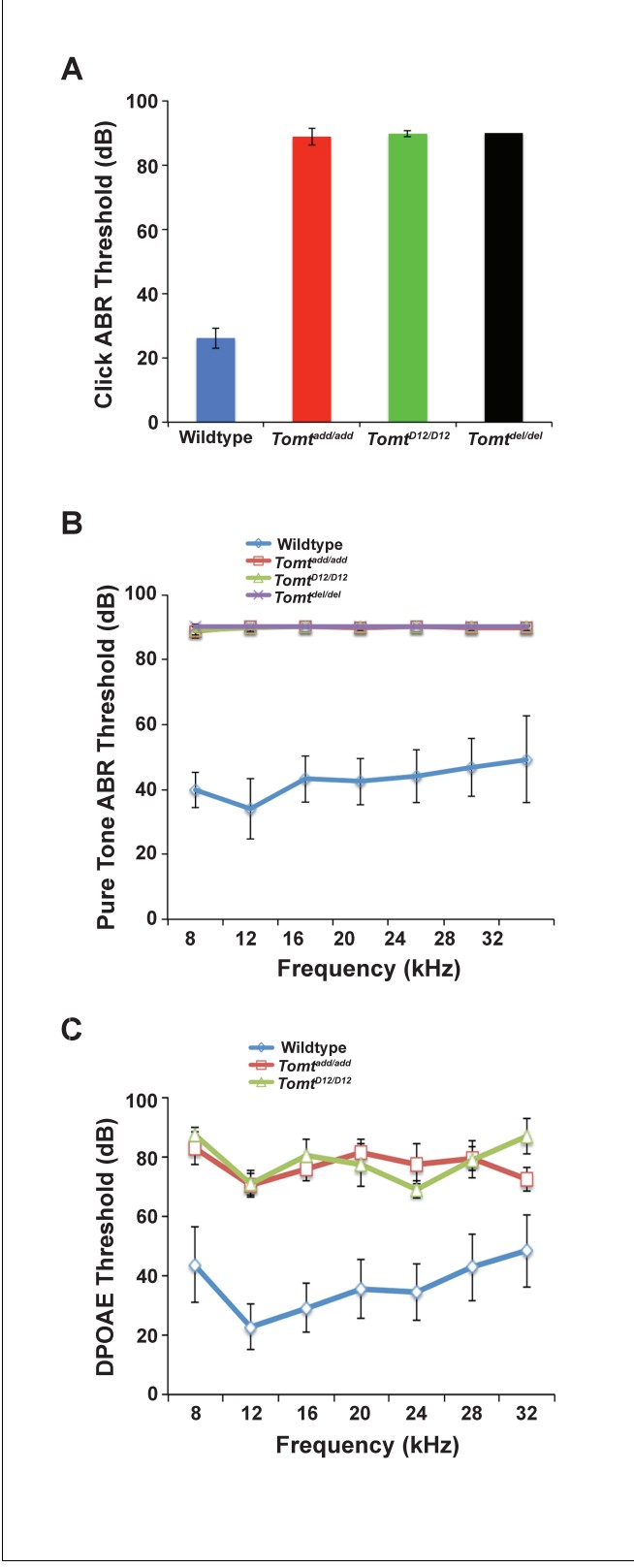

**Figure 2.** Hearing function. (**A**) Auditory brainstem response (ABR) threshold values in response to Click stimuli from 3- to 5-week old wildtype (n = 13), *add* (*Tomt*$^{add/add}$, n = 10), H22_R25del (*Tomt*$^{D12/D12}$, n = 7), and R25Qfs (*Tomt*$^{del/del}$, n = 4). (**B**) ABR threshold values in response to pure tone stimuli of various frequencies for wildtype (n = 13), *add* (*Tomt*$^{add/add}$, n = 10), H22_R25del (*Tomt*$^{D12/D12}$, n = 7), and R25Qfs (*Tomt*$^{del/del}$, n = 4). (**C**) Threshold
*Figure 2 continued on next page*

*Figure 2 continued*

values for distortion product otoacoustic emission (DPOAE) stimuli presented at various frequencies for wildtype
(n = 13), *add* (*Tomt^add/add*, n = 10) and H22_R25del (*Tomt ^D12/D12*, n = 7). All values are mean±SD.

were absent in *Tomt^add/add* and *Tomt^D12/D12* at 4 weeks of age at all frequencies tested (*Figure 2C*).
As these emissions depend on the mechanical activity of OHCs, we conclude that OHC function was
affected in *Tomt* mutant mice.

## Expression and subcellular localization of mTOMT in hair cells

We have previously shown by in situ hybridization that *Tomt* is prominently expressed in cochlear
hair cells (*Du et al., 2008*), suggesting that *Tomt* directly regulates hair cell function. Immunolocali-
zation studies have revealed mTOMT expression in the cytoplasm of hair cells and potentially addi-
tional expression in support cells that was not revealed by in situ hybridization (*Ahmed et al., 2008*).
We made several attempts to raise antibodies to mTOMT but were unable to generate antibodies
of sufficient specificity for immunolocalization studies. To confirm and extend the previous immuno-
localization studies, we therefore generated cDNAs encoding TOMT that were fused in frame at the
N-terminus or C-terminus to sequences encoding an HA-tag (HA-mTOMT and mTOMT-HA)
(*Figure 3A*). We also generated cDNAs encoding murine mTOMT and human LRTOMT2 fused at
the C-terminus to GFP (mTOMT-GFP and LRTOMT2-GFP) (*Figure 3A*). We then expressed the
recombinant proteins by injectoporation (*Xiong et al., 2014*) in organ cultures of the cochlea sen-
sory epithelium at P3 (*Figure 3B*). Protein distribution was analyzed 1 day later by staining with anti-
bodies to HA or examining native GFP fluorescence, combined with phalloidin staining to label
stereocilia (*Figure 3C*). Consistent with previous immunolocalization studies of endogenous proteins
(*Ahmed et al., 2008*) we observed consistently prominent expression of HA-mTOMT, mTOMT-HA
and mTOMT-GFP throughout the cytoplasm of hair cells, but not in their stereocilia (*Figure 3C*, and
data not shown). mTOMT-GFP was not significantly colocalized with GM130, a marker of Golgi, in
heterologous cells or hair cells (*Figure 3—figure supplement 1A,D,E*). In heterologous cells, HA-
mTOMT and mTOMT-GFP partially co-localized with KDEL, a marker of the endoplasmic reticulum
(*Figure 3—figure supplement 1A*) and EEA1 (*Figure 3—figure supplement 1B*). Antibody quality
of KDEL and EEA1 precluded in vivo analysis in injectoporated hair cells. However, it was apparent
that HA-mTOMT and mTOMT-GFP were present but not restricted to the apical cistern, Hensen's
body and the subsurface cistern, locations associated with the endoplasmic reticulum (see
[*Lim, 1986*; *Mammano et al., 1999*]). LRTOMT2-GFP was also localized to the cell body of hair cells,
was not present in the stereocilia (*Figure 3D*) and had a similar distribution in heterologous cells
(*Figure 3—figure supplement 1C*) suggesting that the subcellular localization of mTOMT and
LRTOMT2 is similar and possibly conserved. On rare occasions (<1/10 cells), we observed expression
of HA-mTOMT in stereocilia (data not shown), but the significance of this finding is currently unclear.

Culture conditions might have affected the distribution of recombinant mTOMT in hair cells. To
exclude this possibility, we generated AAV serotype 2/1 vectors expressing mTOMT-HA and
mTOMT-GFP. We then injected AAV-mTOMT-HA and AAV-mTOMT-GFP into the inner ear of mice
at postnatal day (P)0–P1 and determined protein localization at P8 by immunostaining with antibod-
ies to HA or GFP (*Figure 3E*). mTOMT-HA and mTOMT-GFP were prominently detected in the cyto-
plasm of hair cells but not in their stereocilia (*Figure 3F*). While we cannot entirely exclude that the
epitope tags might have affected protein localization, our data are consistent with published immu-
nolocalization studies of endogenous mTOMT protein (*Ahmed et al., 2008*) and suggest that
mTOMT is prominently expressed throughout the cell bodies of hair cells but cannot be detected in
significant amounts in their stereocilia.

## Normal hair bundle morphology in *Tomt* mutants

To define the mechanism by which mTOMT might affect hair cell function, we analyzed the develop-
ment and morphological maturation of the cochlear sensory epithelium in wild-type and *Tomt*-defi-
cient mice. Whole mount staining of the dissected sensory epithelium of *Tomt^add/add*, *Tomt^D12/D12*
and *Tomt^del/del* mice at P5 revealed that the sensory epithelia in mutant mice were patterned into

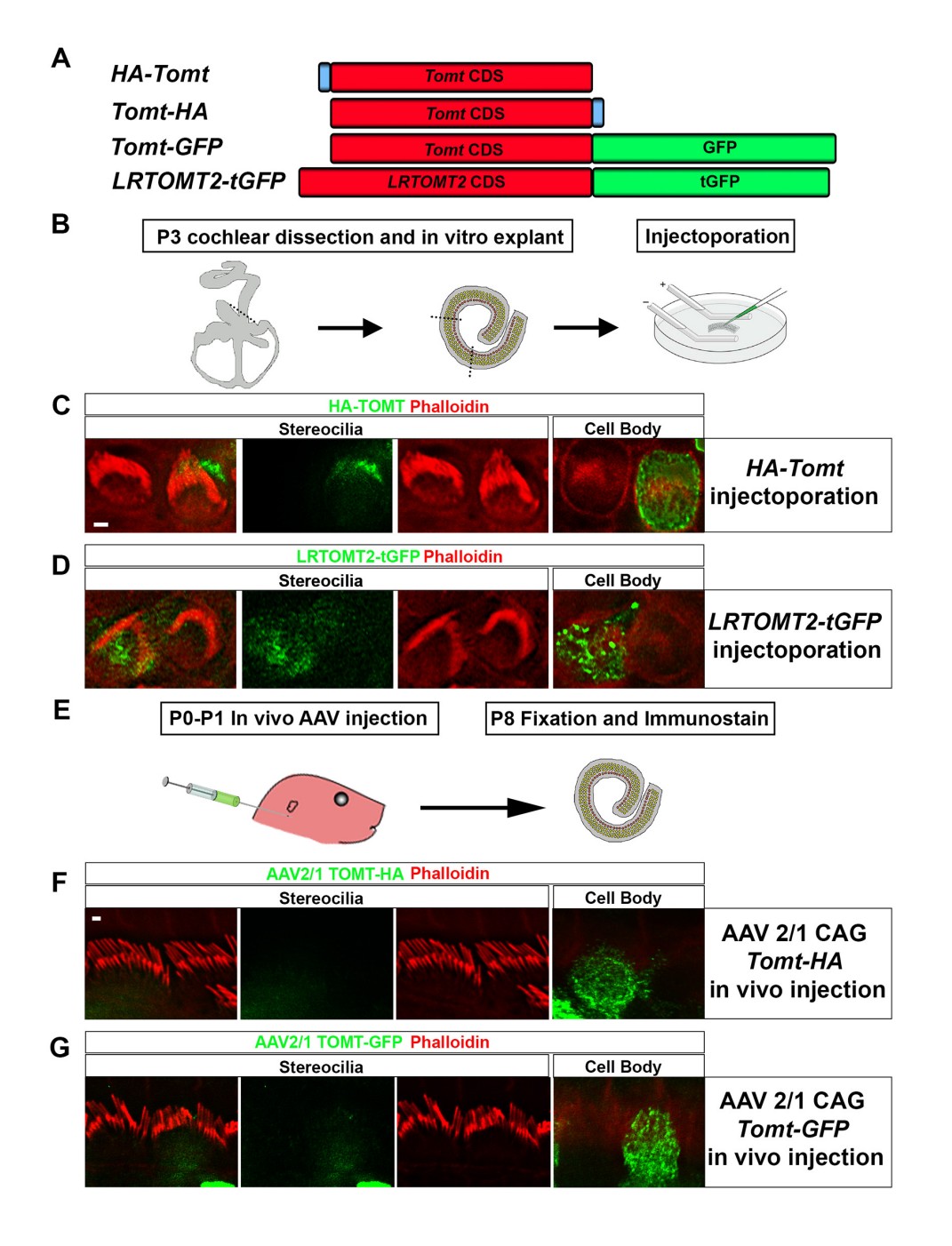

**Figure 3.** Localization of epitope-tagged TOMT in cochlear hair cells. (**A**) Schematic of constructs used for localization studies. Constructs included Mouse TOMT coding sequence tagged at the N-terminus with HA (*HA-Tomt*), C-terminus with HA (*Tomt-HA*), and C-terminus with GFP (*Tomt-GFP*). LRTOMT2, the human homolog of mouse TOMT, was tagged on the C-terminus with tGFP (*LRTOMT2-tGFP*). (**B**) Depiction of experimental paradigm for injectoporation of epitope-tagged constructs (*Xiong et al., 2014*). The Organs of Corti of P3 mice were dissected, sectioned into three parts, and maintained as in vitro explants. Plasmid DNA was microinjected into the space surrounding the basal compartment of hair cells, followed by electroporation. Injectoporated explants were cultured for 24 hr, fixed, and immunostained. (**C**) OHCs injectoporated with HA-TOMT were stained with anti-HA antibody (green) and phalloidin (red) to label actin. Optical sections at the level of the stereocilia and cell body are shown. Note that HA-TOMT localizes to the cell body but not the stereocilia. (**D**) OHCs injectoporated with LRTOMT2-tGFP (green) were stained with phalloidin (red) to label actin. Optical sections at the level of the stereocilia and cell body are shown. Note that LRTOMT2-tGFP localizes to the cell body but not the stereocilia. (**E**) Schematic to describe in vivo inner ear injections of AAV serotype 2/1 viral vectors expressing TOMT-HA or TOMT-GFP from a CAG promoter. At P0-P1, AAV vectors were injected into the inner ear of pups using a Hamilton syringe with a beveled needle tip. One week later, inner ears were fixed and

*Figure 3 continued on next page*

*Figure 3 continued*

immunostained. (**F**) IHCs expressing AAV 2/1 CAG TOMT-HA, stained with anti-HA antibody (green) and phalloidin (red). Optical sections at the level of the stereocilia and cell body are shown. Note that TOMT-HA localizes to the cell body but not the stereocilia. (**G**) IHCs expressing AAV 2/1 CAG TOMT-GFP, stained with anti-GFP antibody (green) and phalloidin (red). Optical sections at the level of the stereocilia and cell body are shown. Note that TOMT-GFP localizes to the cell body but not the stereocilia. Scale bar in (**C**) = 1 µm and applies to (**C,D**). Scale bar in (**F**) = 1 µm, and applies to (**F, G**).

The following figure supplement is available for figure 3:

**Figure supplement 1.** Subcellular localization of epitope-tagged TOMT constructs in heterologous cells and cochlear hair cells.

three rows of OHCs and one row of IHCs with no obvious structural abnormalities (*Figure 4A–C*). At P5, the bundles of outer hair cells (OHCs) in homozygous mutants appeared similar in size to those of wild-type mice and formed a normal staircase pattern (*Figure 4A–D*). For higher resolution studies, we also analyzed hair bundle morphology by scanning electron microscopy (SEM), focusing on *Tomt*$^{add/add}$ mice (*Figure 4D–J*). With SEM, the hair bundles of OHCs had a slightly more rounded morphology, especially in basal regions of the cochlea (*Figure 4E,F,H*). The morphology of inner hair cells (IHCs) was not significantly altered (*Figure 4A–C,I,J*; and data not shown). We thus conclude that mTOMT is not essential for the initial patterning of the cochlear sensory epithelium and for the development of hair cells and their hair bundles. This model is further supported by additional electron microscopic studies and immunohistochemical analyses that are presented below.

## Defects in mechanotransduction in *Tomt* mutants

To further define the mechanisms by which mutations in *Tomt* cause deafness, we determined the extent to which mechanotransduction was affected in *Tomt*-deficient mice. As an initial test, we dissected the cochlear sensory epithelia from control *Tomt*$^{del/+}$ mice and homozygous *Tomt*$^{del/del}$ and *Tomt*$^{add/add}$ mice at P5 and cultured them in the presence or absence of 1 mM Gentamicin, an aminoglycoside antibiotic that enters hair cells through their transduction channels and causes hair cell death (*Figure 5A–D*) (*Alharazneh et al., 2011*; *Kawashima et al., 2011*). The explants were fixed after 24 hr and hair cell survival was evaluated by staining with an antibody to MYO7A as reported (*Figure 5A–D*) (*Alharazneh et al., 2011*). As controls, we also stained sensory epithelia that were cultured in the absence of Gentamicin. Strikingly, while hair-cell death was nearly complete in explants from control mice cultured in the presence but not absence of Gentamicin (*Figure 5A,B*), hair cells were maintained in explants from homozygous *Tomt*$^{del/del}$ and *Tomt*$^{add/add}$ mice even in the presence of Gentamicin (*Figure 5C,D*), suggesting that mechanotransduction-channel function was defective in the mutants.

We next analyzed mechanotransduction by electrophysiology, focusing on hair cells from *Tomt*$^{add/add}$ mice. We stimulated hair bundles of OHCs at P4 and P7 with a stiff glass probe and recorded mechanotransduction currents in the whole-cell configuration (*Figure 5E–G*). As reported (*Grillet et al., 2009*; *Xiong et al., 2012*; *Zhao et al., 2014*), OHCs from control mice had rapidly activating transducer currents, which subsequently adapted (*Figure 5E*). The amplitude of saturated mechanotransduction currents in control OHCs in the mid-apical part of the cochlea at maximal deflection with a stiff glass probe was at 509 ± 50 pA (mean ± SEM) at P4 and 595 ± 56 pA at P7 (*Figure 5F–G*). In contrast, transducer currents were very small (45 ± 6 pA) in P4 hair cells and completely absent in P7 hair cells from *Tomt*$^{add/add}$ mice, even at maximal deflection of their bundles (*Figure 5E–G*). Membrane potential, outward-evoked currents and nonlinear capacitance were normal in OHCs from *Tomt*$^{add/add}$ mice (*Figure 5—figure supplement 1*), suggesting that the mechanotransduction defects were not a secondary consequence of another process affecting the overall health of the hair cells.

## Normal numbers of tip links are present in *Tomt* mutants

Defects in the formation or maintenance of tip links could cause the mechanotransduction defects in *Tomt*-deficient mice. We therefore analyzed hair bundles from control, *Tomt*$^{del/del}$ and *Tomt*$^{add/add}$ mice by immunohistochemistry to analyze the expression of the tip-link components CDH23 and PCDH15 in hair bundles of IHCs (*Figure 6A–D*) and OHCs (*Figure 6—figure supplement 1*). For

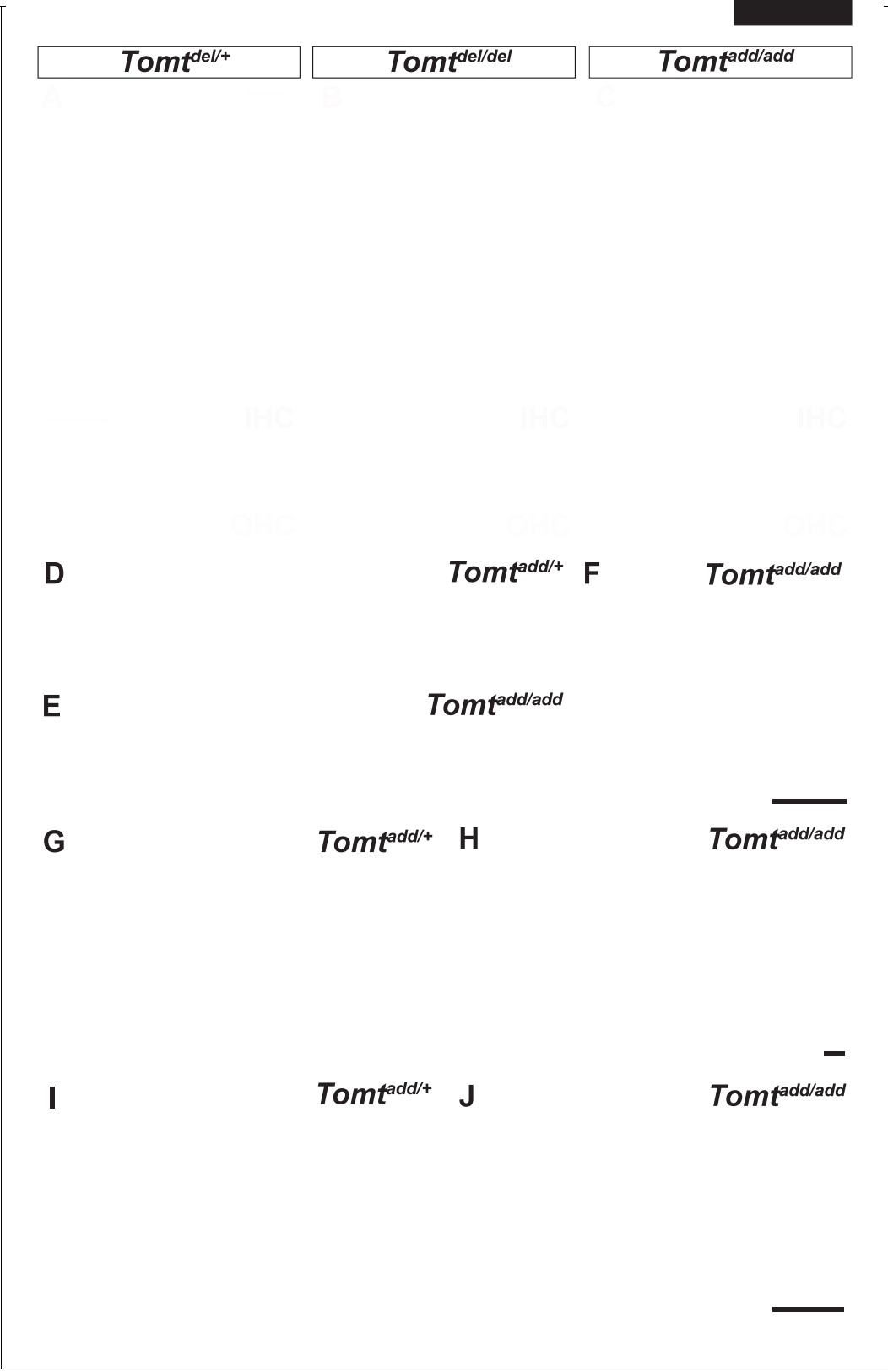

**Figure 4.** Morphology of hair bundles. (A–C) P5 Cochlear whole mounts from the indicated mice stained with phalloidin to label stereocilia. Bottom panels show IHCs and OHCs from each genotype stained for phalloidin. Scale bar in (A) upper panel = 5 µm, applies to upper panels in (A–C). Scale bar in (A) lower panel = 5 µm, applies to lower panels in (A–C). (D–J) SEM analysis of cochlear hair bundles from P8 $Tomt^{add/+}$ and $Tomt^{add/add}$ mice. (D) OHCs from middle region of P8 $Tomt^{add/+}$ cochlea. (E) OHCs from middle region of P8 $Tomt^{add/add}$ cochlea. (F) Example of outer hair cell from

*Figure 4 continued on next page*

*Figure 4 continued*

middle region of P8 *Tomt*[add/add] cochlea. (G) OHCs from base region of P8 *Tomt*[add/+] cochlea. (H) OHCs from base region of P8 *Tomt*[add/add] cochlea. Note the presence of hair bundles with slightly more rounded phenotypes relative to P8 *Tomt*[add/+] cochlea. (I) IHC from middle region of P8 *Tomt*[add/+] cochlea. (J) IHC from middle region of P8 *Tomt*[add/add] cochlea. Scale bar in (E) = 1 μm, applies to (D–E). Scale bar in (F) = 1 μm. Scale bar in (H) = 1 μm, applies to (G,H). Scale bar in (J) = 1 μm, applies to (I,J).

PCDH15, we focused on the CD2 isoform that has been implicated in mechanotransduction (*Pepermans et al., 2014*; *Webb et al., 2011*). We also immunostained for other proteins expressed in hair cells such as PMCA, MYO7A, and Espin and Whirlin (*Figure 6E–L*). We did not observe any difference in the expression or localization of these proteins between wild-type and mutant mice at P5-P8. As predicted, all proteins were expressed within the hair bundle of hair cells (*Figure 6A–L*) with additional expression of MYO7A within the cell body in both wild-type and mutants (*Figure 6G, H*).

To visualize tip links directly, we analyzed hair bundles at P7-P8 by SEM (*Figure 6M,N*) and quantified tip-link numbers in IHCs and OHCs following our previously described procedures (*Xiong et al., 2012*; *Zhao et al., 2014*). There was no statistically significant difference in the number of tip links between wild-type and *Tomt*[add/add] mice (*Figure 6M,N*). We conclude that the mechanotransduction defects in *Tomt*-deficient mice are not primarily caused by defects in tip-link assembly.

## Catecholamines and mechanotransduction

COMT regulates catecholamine levels in the brain and recombinant mTOMT has previously been shown to have catecholamine-O-methyltransferase activity in vitro, albeit significantly less than COMT (*Du et al., 2008*). In the dopamine catabolic pathways, COMT is the rate-limiting enzyme catalyzing the transfer of methyl groups from S-adenosyl methionine (SAM) onto a hydroxyl group of dopamine, thus converting it to 3-methoxytyramine to limit dopamine levels (*Männistö and Kaakkola, 1999*). We reasoned that mutations in mTOMT that affect its enzymatic activity might lead to increased levels of catecholamines in the inner ear that could affect mechanotransduction by an unknown mechanism. To test this model, we first determined the extent to which exogenously provided catecholamines might affect mechanotransduction by cochlear hair cells. We cultured cochlear sensory epithelia from wild-type mice in the presence of high amounts of dopamine, norepinephrine, tolcopone, a widely used COMT inhibitor (*Kiss and Soares-da-Silva, 2014*; *Laatikainen et al., 2013*), and Sinefungin, a global methyltransferase inhibitor (*Borchardt et al., 1979*; *Cheng et al., 2004*) and challenged them with Gentamicin (*Figure 7A*). None of the tested catecholamines or inhibitors prevented Gentamicin-induced hair cell death (*Figure 7B–K*), suggesting that mechanotransduction was not directly affected by these catecholamines or changes in global methyltransferase activity. Analysis of the effects of catecholamines on mechanotransduction was also evaluated by patch-clamp electrophysiology with no noticeable effects on mechanotransduction (*Figure 7—figure supplement 1*).

Next, we analyzed the levels of various catecholamines in the inner ear of wild-type and *Tomt* mutant animals. We dissected the temporal bones of wild-type and *Tomt*[add/add] mice at P5 and analyzed catecholamine levels by high-performance liquid chromatography (HPLC) (*Figure 7L*). We detected norepinephrine, homovanillic acid (HVA) and norepinephrine with no significant difference between wild-type and mutant animals (*Figure 7M*). Serotonin, which like catecholamines is a biogenic monoamine neurotransmitter, was also present with no noticeable difference between wild-type and mutants (*Figure 7M*).

Next, we wanted to test more directly the extent to which methyltransferase activity is important for mTOMT function in hair cells. We took advantage of our *Tomt-GFP* cDNA and introduced a point mutation in a tyrosine residue that is conserved within the methyltransferase domain of mTOMT and mCOMT and critical for enzymatic activity (*Tomt-Y108A-GFP*) (*Figure 7N*) (*Zhang and Klinman, 2011*). We also generated a cDNA containing only the predicted mTOMT methyltransferase domain sequence fused to a C-terminal GFP tag (mTOMT-MT-GFP), and we generated a cDNA encoding a C-terminal GFP fusion with mCOMT (mCOMT-GFP) (*Figure 7N*). The GFP tag on mCOMT has previously been shown not to affect its enzymatic activity (*Sei et al., 2010*). We then tested the extent to which the recombinant cDNAs could rescue defects in mechanotransduction in hair cells from *Tomt-*

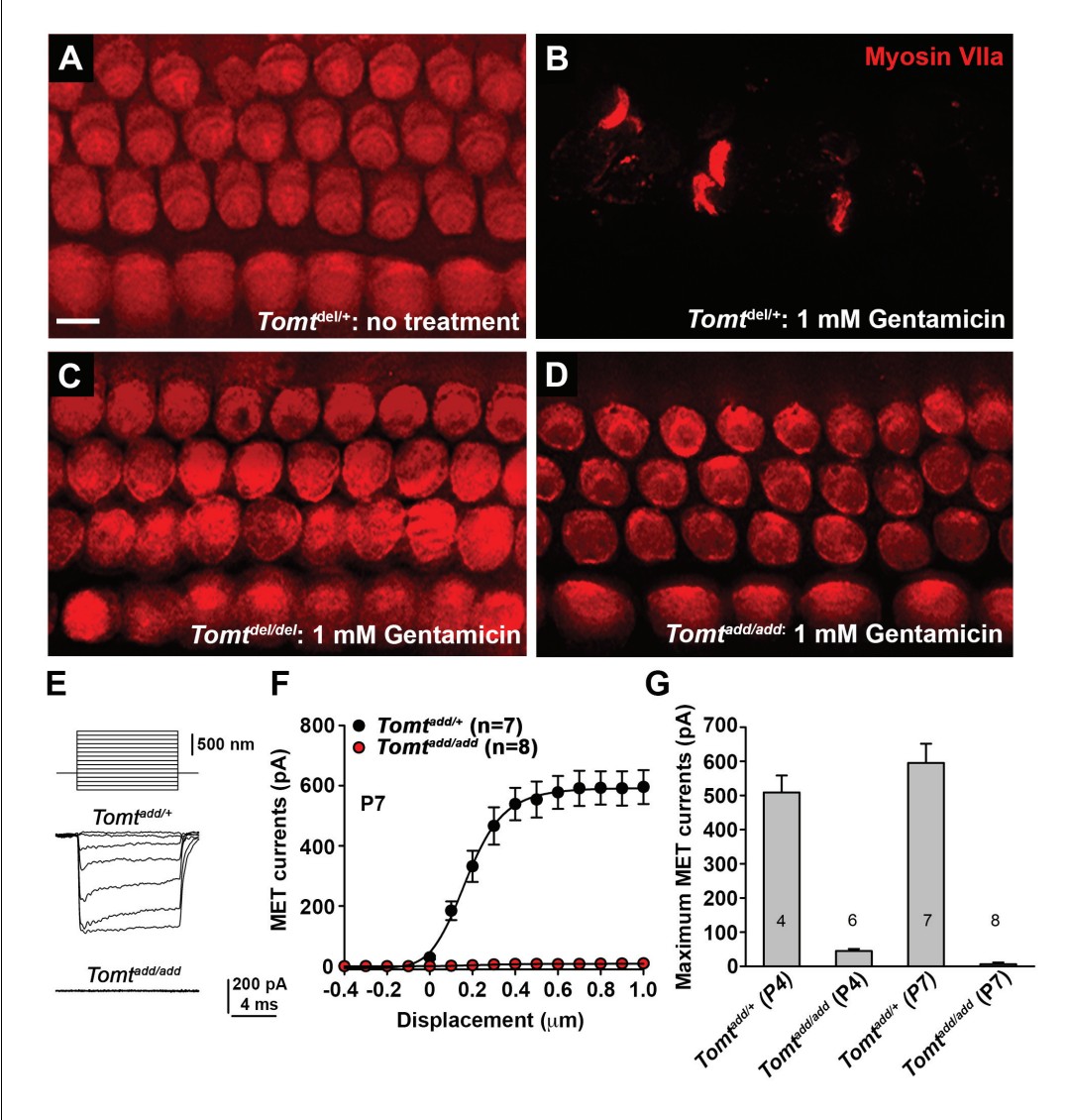

**Figure 5.** Analysis of mechanotransduction in *Tomt* mutants. (A–D) P5 explants from middle region of the cochlea cultured for 24 hr with or without 1 mM Gentamicin, followed by fixation and immunostaining for MYO7A. (A) *Tomt*<sup>del/+</sup> cultured without Gentamicin. (B) *Tomt*<sup>del/+</sup> cultured for 24 hr with 1 mM Gentamicin. Note the almost complete loss of hair cells. (C) *Tomt*<sup>del/del</sup> cultured for 24 hr with 1 mM Gentamicin. Hair cells are spared from Gentamicin-mediated ototoxicity. (D) *Tomt*<sup>add/add</sup> cultured for 24 hr with 1 mM Gentamicin. Hair cells are spared from Gentamicin-mediated ototoxicity. (E) Representative mechanotransduction currents in OHCs from *Tomt*<sup>add/+</sup> and *Tomt*<sup>add/add</sup> mice at P7 in response to a set of 10 ms hair bundle deflections ranging from −400 nm to 1000 nm (100 nm steps). (F) Current displacement plot obtained from similar data as shown in (E). Data are mean ± SEM. (G) Summary of data obtained from experiments as shown in (E–F) for *Tomt*<sup>add/+</sup> and *Tomt*<sup>add/add</sup> at P4 and P7. Data are plotted as maximum MET currents. Number of cells tested is indicated on each column of the graph. Scale bar in (B) = 5 µm and applies to (A–D).

The following figure supplement is available for figure 5:

**Figure supplement 1.** Basic membrane properties of cochlear hair cells in *Tomt* mutants.

deficient mice. Using injectoporation, we expressed the cDNAs in OHCs from *Tomt*<sup>add/add</sup> mice at P3 and analyzed transducer currents after one day in culture (*Figure 7O–Q*). OHCs from the mid-apical region of the cochlea expressing the cDNAs were identified by GFP fluorescence and mechanotransduction currents were evaluated by patch-clamp electrophysiology using a stiff glass probe to deflect hair bundles (*Figure 7O–Q*). Mechanotransduction defects were not rescued by expression of mTOMT-MT-GFP (32 ± 3 pA) or mCOMT-GFP (40 ± 0.5 pA). Unlike TOMT, COMT-GFP localized

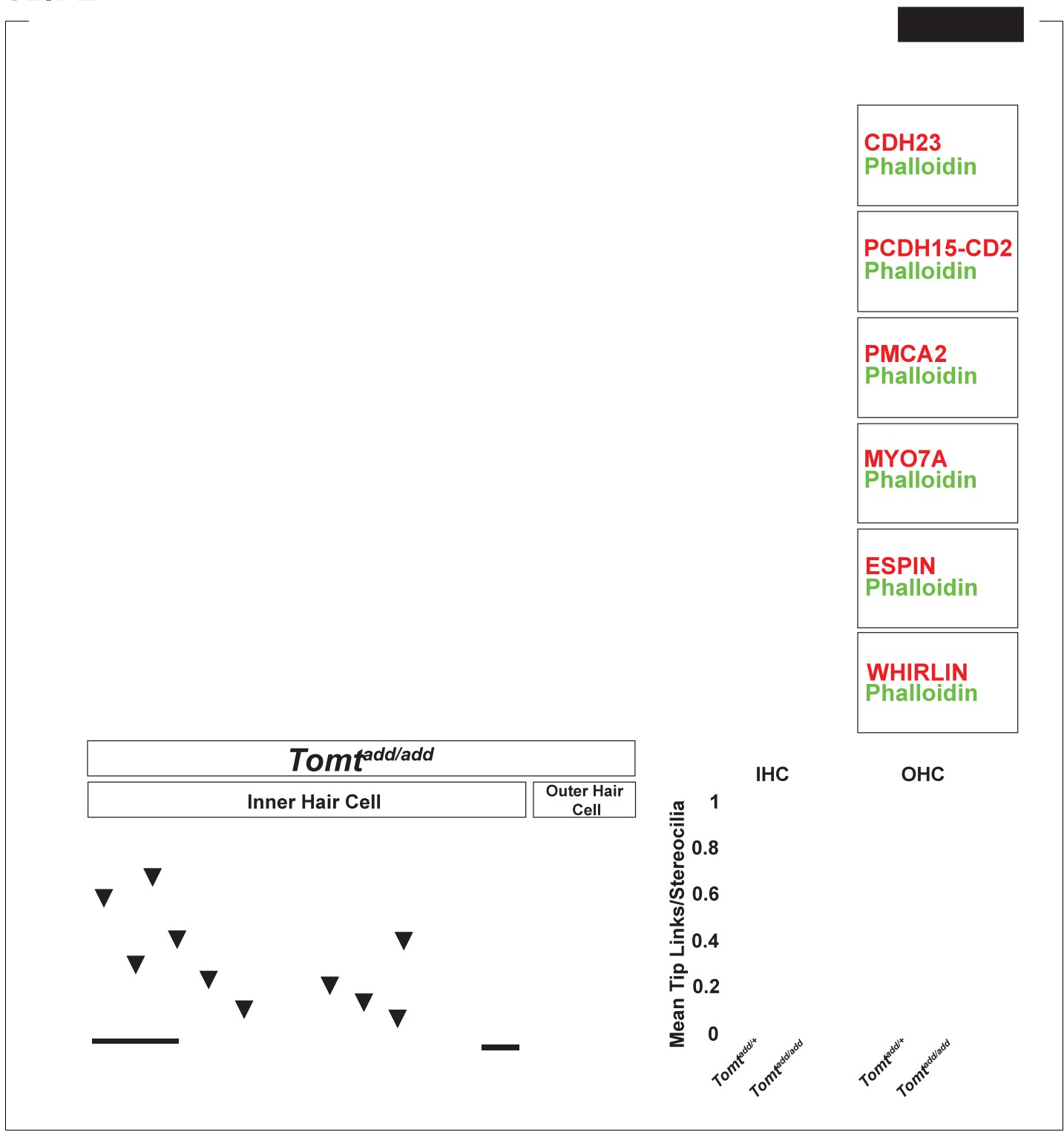

**Figure 6.** Analysis of tip links in *Tomt* mutants. (A,B) P6 cochlear inner hair cells from (A) *Tomt*[add/+] and (B) *Tomt*[add/add] mice stained with anti-Cdh23 (*Siemens et al., 2004*) and phalloidin. (C,D) P7 cochlear inner hair cells from (C) *Tomt*[add/+] and (D) *Tomt*[add/add] mice stained with anti-PCDH15-CD2 (*Webb et al., 2011*) and phalloidin. (E,F) P6 cochlear inner hair cells from (E) *Tomt*[add/+] and (F) *Tomt*[add/add] mice stained with anti-PMCA2 (Abcam) and phalloidin. (G,H) P8 cochlear inner hair cells from (G) *Tomt*[add/+] and (H) *Tomt*[add/add] mice stained with anti-MYO7A (Proteus) and phalloidin. (I,J) P8 cochlear inner hair cells from (I) *Tomt*[add/+] and (J) *Tom* [add/add] mice stained with anti-Espin (BD Biosciences) and phalloidin. (K,L) P7 cochlear inner hair cells from (K) *Tomt*[add/+] and (L) *Tomt*[add/add] mice stained with anti-Whirlin (see Materials and methods) and phalloidin. Scale bar in (A) = 5 μm, and applies to (A–L). (M) Tip links from middle region cochlear hair cells from P7-8 *Tomt*[add/add] mice using SEM. Left and middle panels are from inner hair cell, right panel is from outer hair cell. Left panel scale bar = 500 nm, middle panel scale bar = 100 nm and applies to middle and right panels. (N)
*Figure 6 continued on next page*

*Figure 6 continued*

Quantification of tip-links per stereociliary column from P7-P8 middle region inner and outer hair cells from $Tomt^{add/+}$ (IHC n = 10 cells, OHC n = 9 cells) and $Tomt^{add/add}$ (IHC n = 17 cells, OHC n = 9) mice.
The following figure supplement is available for figure 6:

**Figure supplement 1.** Immunostaining for tip-link components in outer hair cells from *Tomt* mutants.

to stereocilia (*Figure 7—figure supplement 2*) but still could not rescue mechanotransduction defects. In contrast, mTOMT-GFP (231 ± 19 pA) and mTOMT-Y108A-GFP (278 ± 41 pA) rescued transduction to a similar extent. However, rescue of transduction was only partial since transducer currents in OHCs from control mice that were injectoporated with GFP ($Tomt^{add/+}$ + GFP) were at 541 ± 41 pA. Partial rescue of transduction could be a consequence of differences in expression levels of transgenes expressed from heterologous promoters compared to endogenous protein levels in wild-type hair cells or modified function due to epitope tags, but further studies are necessary to address this point (see discussion). Nevertheless, the fact that both mTOMT-GFP and mTOMT-Y108A-GFP rescued transduction to a similar extent, while the methyltransferase mCOMT-GFP or the putative TOMT methyltransferase domain (mTOMT-MT-GFP) could not, suggests that methyltransferase activity is not essential for TOMT function in hair cells.

## Interaction of mTOMT with putative components of the mechanotransduction machinery

To further define the mechanism by which TOMT affects transduction, we tested the hypothesis that mTOMT might interact with putative components of the mechanotransduction channel of hair cells (TMC1, TMC2, TMIE, TMHS/LHFPL5) or with tip-link proteins (PCDH15, CDH23). We therefore co-expressed TOMT carrying either a C-terminal HA-tag (TOMT-HA) or a C-terminal FLAG tag (TOMT-FLAG) together with TMC1, TMC2, TMIE and TMHS/LHFPL5 in HEK293 cells. TMC1 and TMC2 contained N-terminal Myc tags, TMIE carried a C-terminal HA tag, and LHFPL5/TMHS was tagged at the N-terminus with HA. We also co-expressed TOMT-FLAG with the tip-link cadherins CDH23 and PCDH15. Significantly, epitope-tagged versions of TOMT (TOMT-FLAG or TOMT-HA) could be co-immunoprecipitated with Myc-TMC1, Myc-TMC2, TMIE-HA, HA-TMHS/LHFPL5 and PCDH15, but not with CDH23 (*Figure 8A–E*, data not shown).

We could not detect TOMT in stereocilia of hair cells (*Figure 3*) and thus hypothesized that TOMT might transiently interact with components of the mechanotransduction complex to facilitate protein transport into stereocilia. To test this model, we analyzed protein distribution in hair cells from wild-type and TOMT-deficient animals. Our analysis of tip-link cadherins (*Figure 6*) already suggested that TOMT was not essential for PCDH15 transport. To evaluate distribution of TMHS/LHFPL5, we carried out immunolocalization studies in hair cells from wild-types and mutants but did not observe any obvious effect of TOMT on the distribution of TMHS/LHFPL5 (*Figure 8F*). While antibodies against TMIE and TMC1/2 have previously been described, their sensitivity is very low (*Kurima et al., 2015*; *Zhao et al., 2014*). Thus, to evaluate effects of TOMT on the distribution of TMIE and TMC1/2 in hair cells, we expressed epitope-tagged versions of the proteins by injectoporation in OHCs from wild-type and *Tomt*-deficient animals (*Figure 8G–I*). TMIE-HA was effectively targeted to the hair bundles of wild-type and *Tomt*-deficient animals (*Figure 8F*). Myc-TMC1 and Myc-TMC2 were both transported into the stereocilia of hair cells from control animals, although Myc-TMC2 was consistently observed at much higher levels compared to Myc-TMC1. Strikingly, both Myc-TMC1 and Myc-TMC2 were absent from the stereocilia of *Tomt*-deficient hair cells (*Figure 8H,I,K*). When mTOMT-GFP was coexpressed with Myc-Tmc2 in *Tomt*-deficient hair cells, Myc-Tmc2 stereociliary localization was rescued (*Figure 8J,K*). These findings suggest that in hair cells, TOMT is essential to regulate the transport of TMC1 and TMC2 into stereocilia.

Notably, previous studies have shown that in heterologous cells such as HEK293 cells or COS-7 cells, TMC1 and TMC2 cannot be effectively expressed at the cell surface; instead, the proteins appear to be retained intracellularly, likely within the endoplasmic reticulum (*Beurg et al., 2015*; *Labay et al., 2010*). Co-expression of TOMT-GFP or HA-TOMT did not alter the distribution of

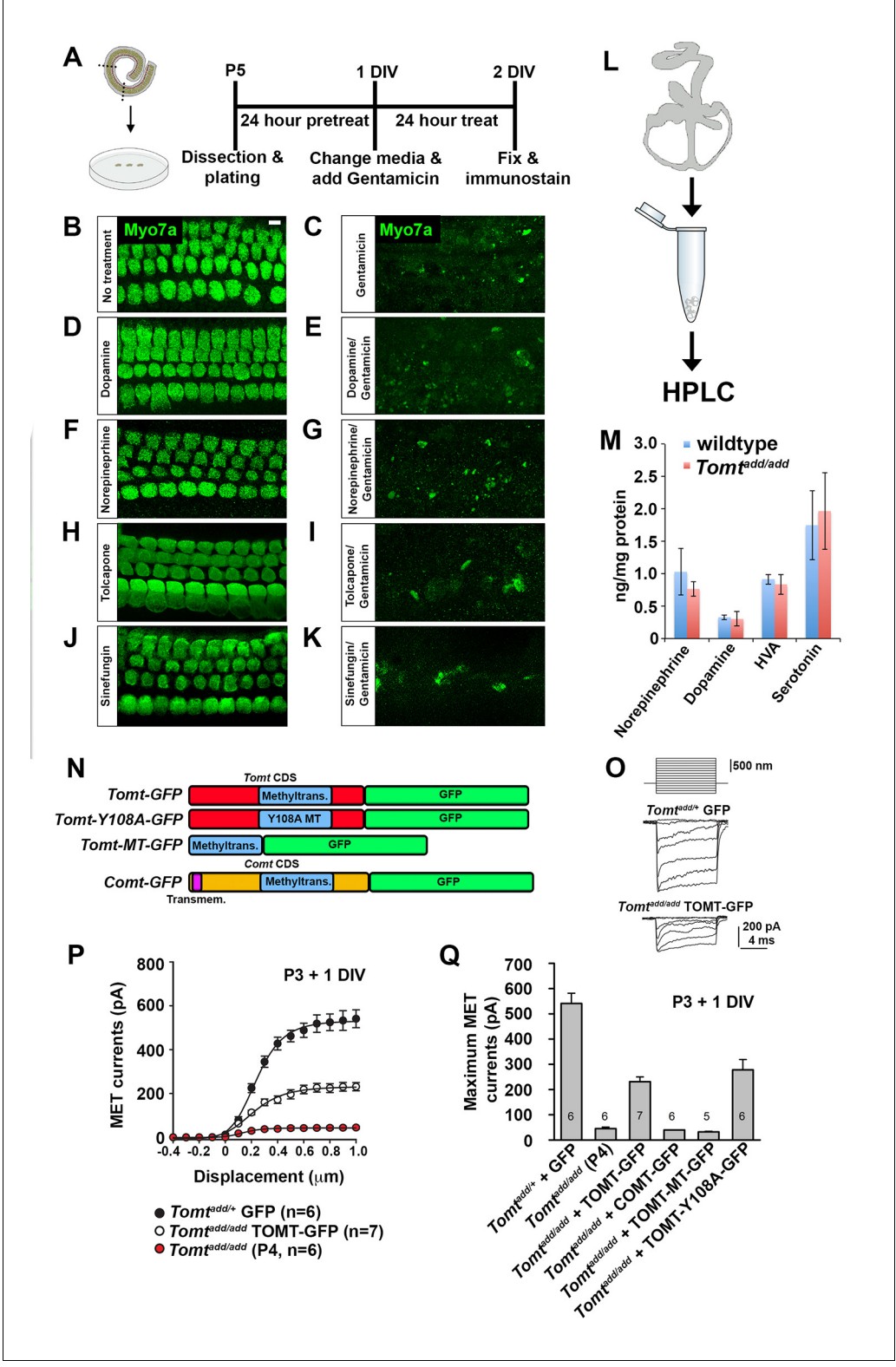

**Figure 7.** Effects of catecholamines on mechanotransduction. (**A**) Experimental paradigm for assaying the effect of catecholamines on Gentamicin-mediated ototoxicity. The organ of Corti of P5 C57BL/6J mice was dissected, sectioned into three parts, and plated for in vitro culture. Explants were pretreated with catecholamines or inhibitors for 24 hr. Media was replaced with fresh media containing catecholamines with or without 1 mM Gentamicin. Explants were cultured for 24 additional hours, followed by fixation and immunostaining for MYO7A
*Figure 7 continued on next page*

*Figure 7 continued*

(green) to reveal hair cells. (**B**) Untreated P5 explants cultured without catecholamine or Gentamicin. (**C**) P5 explants cultured with Gentamicin. Note the significant loss of hair cells. (**D**) P5 explants treated for 48 hr with 2.5 µM Dopamine, with a media change with fresh Dopamine at 24 hr. (**E**) P5 explants pretreated for 24 hr with 2.5 µM Dopamine, followed by 24 hr of 2.5 µM Dopamine and 1 mM Gentamicin. Dopamine treatment does not rescue cells from Gentamicin-mediated ototoxicity. (**F**) P5 explants treated for 48 hr with 1 µM Norepinephrine, with a media change with fresh Norepinephrine at 24 hr. (**G**) P5 explants pretreated for 24 hr with 1 µM Norepinephrine, followed by 24 hr of 1 µM Norepinephrine and 1 mM Gentamicin. Norepinephrine treatment does not rescue cells from Gentamicin-mediated ototoxicity. (**H**) P5 explants treated for 48 hr with 10 µM Tolcapone, a COMT inhibitor, with a media change with fresh Tolcapone at 24 hr. (**I**) P5 explants pretreated for 24 hr with 10 µM Tolcapone, followed by 24 hr of 10 µM Tolcapone and 1 mM Gentamicin. Tolcapone treatment does not rescue cells from Gentamicin-mediated ototoxicity. (**J**) P5 explants treated for 48 hr with 1 µM Sinefungin, a global methyltransferase inhibitor, with a media change with fresh Sinefungin at 24 hr. (**K**) P5 explants pretreated for 24 hr with 1 µM Sinefungin, followed by 24 hr of 1 µM Sinefungin and 1 mM Gentamicin. Sinefungin treatment does not rescue cells from Gentamicin-mediated ototoxicity. Scale bar in (**B**) = 5 µm, applies to (**B–K**). (**L**) Experimental paradigm for assaying catecholamine levels of inner ears from P5 wildtype (C57BL/6J) and *Tomt*$^{add/add}$ using HPLC. Inner ears from P5 wildtype and *Tomt*$^{add/add}$ mice were dissected and snap frozen on dry ice. Each sample contained two temporal bones pooled together and represented one animal. Catecholamine levels were assayed using HPLC and normalized to total protein content. (**M**) Quantification of catecholamine levels. Norepinephrine, Adrenaline, DOPAC, Dopamine, 5-HIAA, HVA, Serotonin, and 3-MT were assayed, but only Norepinephrine, Dopamine, HVA, and Serotonin were detected. Absolute levels were normalized to total protein. Values represent mean ± SD of C57BL/6 (n = 5 animals, 10 temporal bones) or *Tomt*$^{add/add}$ (n = 5 animals, 10 temporal bones). (**N**) Schematic for constructs used for injectoporation experiments. Constructs included mouse TOMT coding sequence (CDS) tagged at the C terminus with GFP (*Tomt-GFP*), mouse TOMT CDS with Y108A mutation (orthologous conserved amino acid critical for COMT enzymatic activity; [*Zhang and Klinman, 2011*]) tagged at the C-terminus with GFP (*Tomt-Y108A-GFP*), mouse TOMT methyltransferase (MT) domain (based on NCBI conserved protein domain family cl17173) tagged at the C-terminus with GFP (*Tomt-MT-GFP*), and mouse COMT CDS tagged at the C-terminus with GFP (*Comt-GFP*). Methyltransferase and transmembrane domains indicated. (**O**) Representative mechanotransduction currents in P3+1 DIV OHCs from *Tomt*$^{add/+}$ injectoporated at P3 with GFP and *Tomt*$^{add/add}$ injectoporated at P3 with TOMT-GFP in response to a set of 10 ms hair bundle deflections ranging from −400 nm to 1000 nm (100 nm steps). (**P**) Current displacement plots obtained from similar data as shown in O. (**Q**) Plot of maximum MET currents obtained from similar data as in (**O–P**) from OHCs injectoporated with constructs described in (**N**). Data are mean ± SEM.

The following figure supplements are available for figure 7:

**Figure supplement 1.** Effects of catecholamine manipulation on mechanotransduction.

**Figure supplement 2.** mCOMT-GFP expression in cochlear hair cells.

---

TMC1 and TMC2 in COS-7 cells (*Figure 8—figure supplement 1*, and data not shown), suggesting that TOMT alone is not sufficient to regulate the transport of TMC1 and TMC2 to the cell membrane.

## Effect of mTOMT on reverse polarity currents

During the developmental maturation of hair cells, their hair bundles are less directionally sensitive and transducer currents can initially be evoked by deflection of the hair bundle in the opposite from normal direction (*Kim et al., 2013*; *Kindt et al., 2012*; *Marcotti et al., 2014*; *Waguespack et al., 2007*). Similar reverse polarity currents can be evoked in hair cells lacking tip links (*Alagramam et al., 2011*; *Kim et al., 2013*; *Marcotti et al., 2014*) and in hair cells from mice carrying mutations in the genes encoding TMHS/LHFPL5, TMIE, TMC1/2 and MyoXVa (*Beurg et al., 2015*; *Kim et al., 2013*; *Stepanyan and Frolenkov, 2009*; *Zhao et al., 2014*). Thus, reverse polarity currents have so far been observed consistently in hair cells from mice with mutations that affect mechanotransduction. These reverse-polarity currents depend on PIEZO2 ion channels that are localized at the base of stereocilia (*Wu et al., 2017*). We wondered whether reverse polarity currents could be evoked in hair bundles from *Tomt*-deficient mice. For this purpose, we used the fluid-jet stimulation system that has previously been used to deflect hair bundles in the normal-polarity and

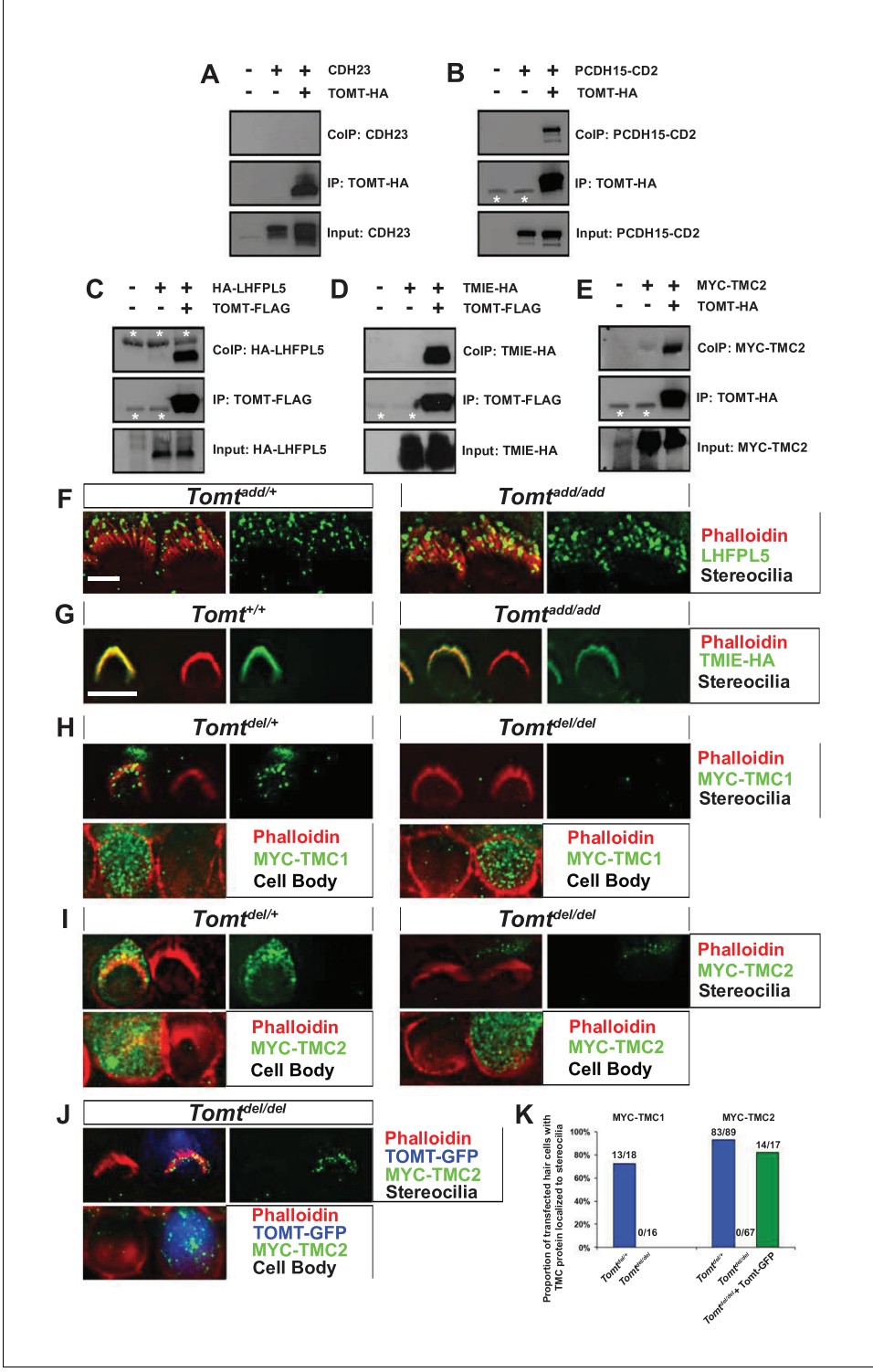

**Figure 8.** Protein-protein interactions with mTOMT. (**A–E**) HEK293 cells were transfected with the constructs indicated on top of each panel to perform co-immunoprecipitation (CoIP) experiments. Immunoprecipitations were carried out with HA (**A,B,E**) or Flag (**C,D**) antibodies, followed by Western blotting to detect proteins. The upper panels show CoIP results, the middle row shows IP results and the lower rows show input protein. White asterisks indicate 25 kDa light-chain IgG bands from antibodies used for IP. (**F**) P6 whole mount cochleas from *Tomt*$^{add/+}$ (left panels) and *Tomt*$^{add/add}$ mice (right panels) stained with anti-TMHS/LHFPL5 (*Xiong et al., 2012*) and phalloidin. Scale bar in (**F**) = 5 μm, applies to both panels. (**G**) OHCs from P3 wild-type (C57BL/6J) (left) and *Tomt*$^{add/add}$ (right) mice were injectoporated with TMIE-HA, cultured for 24 hr, fixed and stained with anti-HA

*Figure 8 continued on next page*

*Figure 8 continued*

antibody and phalloidin. TMIE-HA localizes to the hair bundle of both wildtype and *Tomt*<sup>add/add</sup> OHCs. (**H**) Outer hair cells from P3 *Tomt*<sup>del/+</sup> (left) and *Tomt*<sup>del/del</sup> (right) mice were injectoporated with MYC-TMC1, cultured for 24 hr, fixed and stained with anti-MYC antibody and phalloidin. MYC-TMC1 localizes to the hair bundle of *Tomt*<sup>del/+</sup> but is absent from the hair bundle of *Tomt*<sup>del/del</sup> OHCs. Optical sections at the level of the hair bundle and cell body are shown. (**I**) Outer hair cells from P3 *Tomt*<sup>del/+</sup> (left) and *Tomt*<sup>del/del</sup> (right) mice were injectoporated with MYC-TMC2, cultured for 24 hr, fixed and stained with anti-MYC antibody and phalloidin. MYC-TMC2 localizes to the hair bundle of *Tomt*<sup>del/+</sup> but is absent from the hair bundle of *Tomt*<sup>del/del</sup> OHCs. Optical sections at the level of the hair bundle and cell body are shown. (**J**) Outer hair cells from P3 *Tomt*<sup>del/del</sup> mice were injectoporated with mTOMT-GFP and MYC-TMC2, cultured for 24 hr, fixed and stained with anti-MYC antibody and phalloidin. MYC-TMC2 localizes to the hair bundle of *Tomt*<sup>del/del</sup> hair cells in the presence of exogenous mTOMT-GFP. Optical sections at the level of the hair bundle and cell body are shown. (**K**) Quantification of injectoporation experiments shown in (**H–J**). The data are plotted as the proportion of injectoporated outer hair cells for each genotype and construct that exhibited TMC protein localized to stereocilia. The proportion of cells for each condition are indicated on the columns. Only healthy transfected hair cells, as determined by stereocilia morphology, were included for analysis. Scale bar in (**G**) = 5 µm, applies to (**G–J**).

The following figure supplement is available for figure 8:

**Figure supplement 1.** Analysis of protein localization in heterologous cells.

---

reverse-polarity direction (*Beurg et al., 2016*; *Kim et al., 2013*; *Zhao et al., 2014*). We recorded regular normal-polarity currents in hair cells from P4 control *Tomt*<sup>add/+</sup> mice (*Figure 9A,B*). Small normal-polarity currents were also observed in P4 *Tomt*<sup>add/add</sup> mutants (*Figure 9A,B*), similar to results obtained when hair cells were stimulated at P4 with a stiff glass probe (*Figures 5* and *7*). However, unlike OHCs from any other mutant mouse strain with defects in mechanotransduction, we did not observe reverse polarity currents in OHCs from *Tomt*<sup>add/add</sup> mutants (*Figure 9A*). Surprisingly, reverse polarity currents of similar amplitude could be evoked in hair cells from controls and *Tomt*<sup>add/add</sup> mutants after treatment with BAPTA (*Figure 9A,C*). This result is striking and suggests that loss of TMC stereociliary localization alone is not sufficient to induce reverse polarity currents and that mTOMT is essential for the process of inducing reverse polarity currents in TMC mutant animals. Thus, mTOMT has a critical role in the generation of reverse polarity currents downstream of TMC (see Discussion).

## Discussion

The study of the tip-link cadherins and their associated proteins has revealed an intriguing asymmetry in the mechanotransduction complex of hair cells. CDH23 homodimers form the upper part of tip links and their cytoplasmic domains bind to protein-complexes that contain harmonin, SANS and MYO7A (*Figure 1A*). PCDH15 homodimers form the lower part of tip links and can interact with TMHS/LHFPL5, TMIE, and TMC1/2 (*Figure 1A*). The mechanisms that lead to the assembly of the asymmetric tip-link mechanotransduction complex are largely unknown. Here, we show that mTOMT has a critical function in this process by regulating the distribution of TMC1/2 proteins in stereocilia. This finding was unexpected since based on sequence homology between mTOMT and COMT it was initially proposed that mTOMT might affect hair cell function by the regulation of catecholamine levels in the inner ear (*Du et al., 2008*). However, the findings presented here suggest that defects in the assembly and transport of the mechanotransduction machinery of hair cells are likely intricately linked to the mechanism by which mutations in *Tomt* in mice and *LRTOMT* in humans cause deafness. This process appears to be largely independent of the function of TOMT as a methyltransferase. Our findings do not exclude that mTOMT has additional functions in auditory hair cells that depend on its methyltransferase activity that have not been detected by our analysis.

The fact that proteins are precisely confined to specific domains within the tip-link complex makes it unlikely that proteins reach their position purely by random diffusion. It seems more likely that specific transport and retention mechanisms shape protein complexes at tip links. As one model, proteins that are targeted to the upper and lower end of tip links might be part of different transport complexes. In this model, CDH23 and its associated proteins harmonin, SANS and MYO7A might be

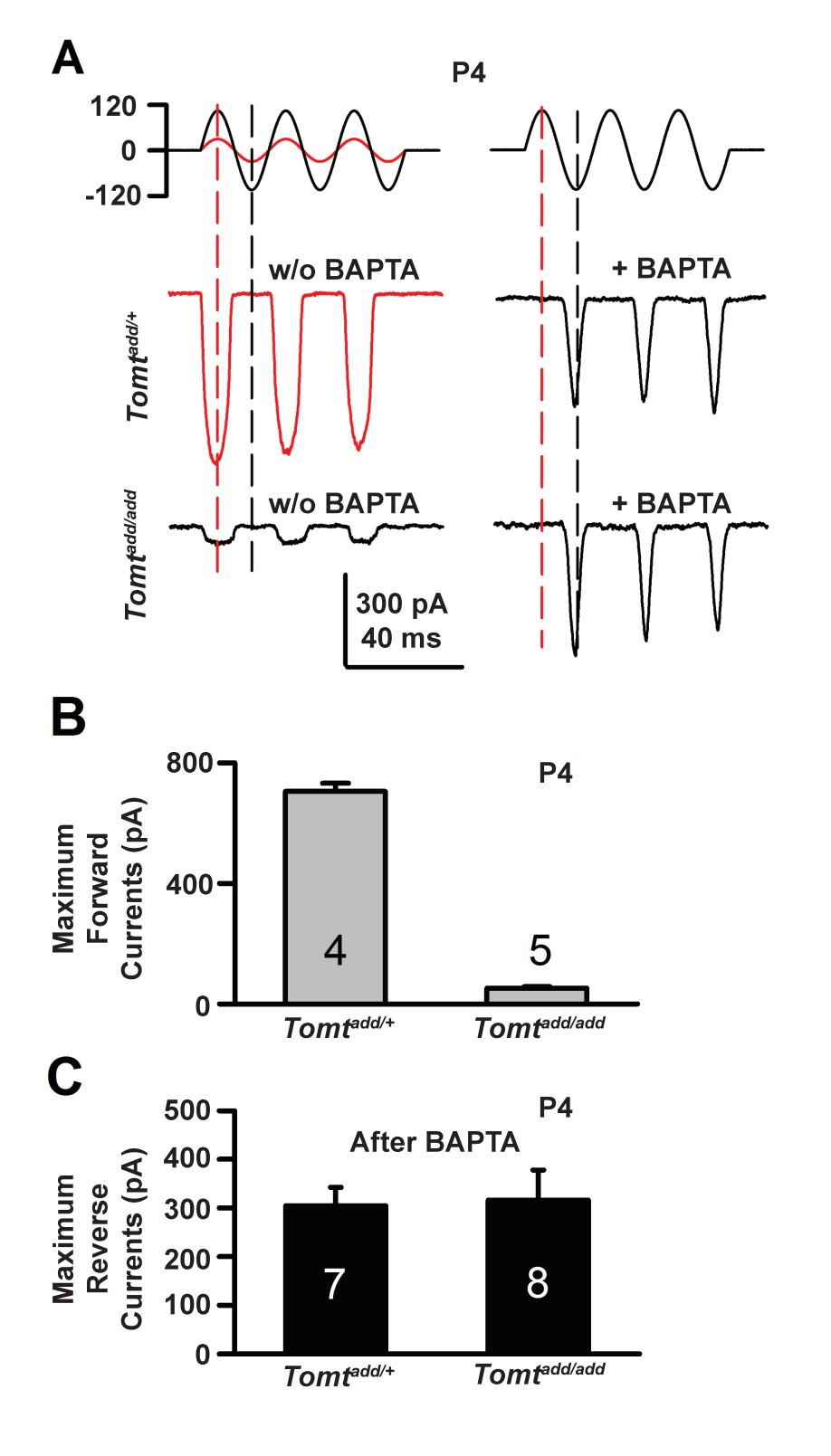

**Figure 9.** Forward and reverse polarity currents in *Tomt* mutants. (**A**) Representative mechanotransduction currents in response to sinusoidal deflection of hair bundles at P4 for OHCs from *Tomt^add/+* and *Tomt^add/add* mice with and without 5 mM BAPTA treatment. All recordings were from middle or apical OHCs at a holding potential of −70 mV. The stimulus monitor (the driving voltage to the fluid jet) is shown at the top. A lower voltage (red trace) was necessary to obtain similar displacements in *Tomt^add/+* mice relative to *Tomt^add/add* mice (black trace). Positive driving voltage denotes
*Figure 9 continued on next page*

*Figure 9 continued*

displacement toward the tallest edge of the hair bundle. (B) Plot of maximum forward MET currents obtained from similar data as A. Number of cells for each condition is indicated on graph. (C) Plot of maximum reverse currents obtained from similar data as in (A). Number of cells for each condition is indicated on graph.

pre-assembled into a multimeric complex within vesicles at the base of stereocilia. The vesicles might then fuse at the base of stereocilia with the plasma membrane, leading to insertion of CDH23 into the membrane and exposing its cytoplasmic interaction partners to the cytoskeleton of stereocilia, thus allowing for MYO7A-driven transport along the actin filaments. Similarly, PCDH15 might be pre-assembled with TMHS/LHFPL5, TMIE and TMC1/2 into complexes that travel together into stereocilia, possibly also by a myosin motor driven process. However, experimental evidence suggests that this model may be too simplistic. This has been most clearly demonstrated for proteins that interact with CDH23 at the upper end of tip links such as harmonin, and SANS. MYO7A has been implicated in the transport of harmonin and SANS to the UTLD (*Bahloul et al., 2010*; *Boëda et al., 2002*; *Senften et al., 2006*), but CDH23 transport occurs in the absence of MYO7A (*Senften et al., 2006*). CDH23 interacts with MYO1C (*Siemens et al., 2004*), which is an alternative candidate protein to mediate CDH23 transport. It is currently unclear why different components of the protein complex at the upper tip-link end are transported by distinct mechanisms. As one possibility, different transport mechanisms might facilitate repair processes. Harmonin, SANS, and MYO7A are clustered at the electron dense UTLD (*Grati and Kachar, 2011*; *Grillet et al., 2009*), which might be a stable structure. CDH23 replacement following damage might be possible without disassembly of the UTLD thus accelerating repair. Further studies will be necessary to test this model.

Our previous studies and the findings presented here suggest that proteins at the lower end of tip links are similarly targeted to their location by distinct mechanisms. The myosin motor proteins (if any) that might be involved in the transport of PCDH15, TMHS/LHFPL5, TMIE, and TMC1/2 still need to be defined, although MYO7A can bind to PCDH15 and might participate in PCDH15 transport (*Senften et al., 2006*). However, protein-protein interactions alone reveal little about transport mechanisms. For example, TMHS/LHFPL5 binds to PCDH15 and TMIE, but it does not bind to TMC1/2 (*Beurg et al., 2015*; *Xiong et al., 2012*; *Zhao et al., 2014*); only PCDH15 depends on TMHS/LHFPL5 for localization to the tip-link complex (*Xiong et al., 2012*). Tip-link localization of TMC1, which does not bind to TMHS/LHFPL5, depends on TMHS/LHFPL5 (*Beurg et al., 2015*). Overall, the studies suggest that while PCDH15 can bind to TMHS/LHFPL5, TMIE, and TMC1/2, assembly of the entire protein complex is not critical for transport of all proteins into stereocilia. This concept extends to the findings reported here for mTOMT. While mTOMT can bind to several components of the transduction complex, it is essential only for the transport of TMC1/2. The data do not exclude that under normal conditions the proteins are transported together into stereocilia. However, even without full assembly of the protein complex due to mutations, some of the remaining proteins can still be effectively transported although it is conceivable that the rate of transport is somewhat reduced.

mTOMT differs significantly from other proteins implicated in shuttling components of the tip-link complex into stereocilia since mTOMT, unlike TMHS/LHFPL5 and MYO7A, cannot be detected within stereocilia. Based on the localization of mTOMT in the cytoplasm, it seems more likely that mTOMT acts in a transient transport compartment prior to the insertion of proteins into the membrane of stereocilia. Expression and localization studies in heterologous cells are consistent with this model.

We and others have shown that TMC1/2 proteins are retained in an intracellular compartment of heterologous cells that were transfected to express TMC1/2 (*Figure 8—figure supplement 1*, [*Beurg et al., 2015*; *Labay et al., 2010*]). We show here that mTOMT that is co-expressed with TMC1/2 is not sufficient for insertion of TMC1/2 into the cell membrane of heterologous cells. Other proteins including TMHS/LHFPL5 may cooperate with mTOMT in protein transport either in concert or possibly acting in a sequential manner in different subcellular compartments. Finally, while our data show that TMC1 and TMC2 when overexpressed do not travel effectively into the hair bundles of TOMT-deficient hair cells, TOMT might affect endogenous TMC1/2 also in other ways, for example by affecting protein levels of the two transmembrane proteins.

The data presented here are consistent with a model where mTOMT acts independently of its enzymatic function in regulating the localization of TMC1/2, possibly acting as an adaptor or shuttle protein. In support of this model, mTOMT binds to PCDH15, TMIE, TMHS/LHFPL5 and TMC1/2. Expression of the methyltransferase domain of mTOMT cannot rescue the mechanotransduction defect of mTOMT-deficient hair cells, while mTOMT carrying a mutation that is predicted to abolish methyltransferase activity still rescues transduction. It should be noted that methyltransferase activity of wild-type recombinant mTOMT when compared to mCOMT appears to be low, at least for prevalent substrates of mCOMT such as norepinephrine in vitro (*Du et al., 2008*). However, mTOMT might have other substrates that are metabolized more efficiently than norepinephrine that are important for auditory function. For example, the catecholamine dopamine has been implicated in the regulation of auditory function by pharmacological and immunohistochemical evidence. Efferent neurons form synapses with OHCs and with the nerve endings of afferents that innervate IHCs (*Eybalin, 1993*; *Puel, 1995*). A small number of efferent fibers have been shown to be dopaminergic, and dopamine can modulate the activity of afferent neurons that synapse on IHCs (*Darrow et al., 2006*). Dopamine can also modulate the activity of afferent neurons that synapse on IHCs (*Eybalin, 1993*; *Puel, 1995*). However, we note that there were no differences in Dopamine levels in the inner ear of $Tomt^{add}$ mutants, and exogenous treatment with Dopamine did not affect mechanotransduction (*Figure 7*). Intriguingly, catechol-o-methyltransferase activity is not completely abolished in the brain of mice with mutations affecting mCOMT (*Gogos et al., 1998*). It would thus be interesting to test the extent to which mTOMT is responsible for the remaining methyltransferase activity, for example by analyzing mice deficient for both mCOMT and mTOMT, and if so, which are endogenous substrates of mTOMT.

During the developmental maturation of hair cells, hair bundles slowly acquire directional sensitivity. Initially, transducer currents can be evoked even by deflection of the hair bundle in the opposite from normal direction (*Beurg et al., 2016*; *Kim et al., 2013*; *Kindt et al., 2012*; *Marcotti et al., 2014*; *Stepanyan and Frolenkov, 2009*; *Waguespack et al., 2007*; *Wu et al., 2017*). Reverse polarity currents have also been consistently observed in mice with mutations that affect proteins of the complex that regulate transduction (*Beurg et al., 2015*; *Kim et al., 2013*; *Stepanyan and Frolenkov, 2009*; *Zhao et al., 2014*). These reverse polarity currents depend on the PIEZO2 ion channel that is concentrated near the base of stereocilia (*Wu et al., 2017*). There has been a striking correlation between the loss of normal polarity currents and the emergence of reverse polarity currents (*Beurg et al., 2015*; *Kim et al., 2013*; *Zhao et al., 2014*). We were therefore surprised that both normal and reverse polarity currents are affected by mTOMT. How can these data be reconciled? Since reverse polarity currents can be evoked in hair cells lacking TMC1/2, the loss of these proteins from stereocilia, as observed in mTOMT-deficient hair cells, is alone insufficient to unmask reverse polarity currents. Instead, the data suggest that mTOMT might act in a molecular pathway with TMC1/2 to regulate the emergence of reverse polarity currents. Our previous studies suggest that reverse polarity currents are regulated by the intracellular calcium concentration that is increased in response to disruption of the mechanotranduction machinery of hair cells (*Wu et al., 2017*). Perhaps, mTOMT has a role in regulating intracellular calcium levels for example by regulating not only the distribution and activity of TMC1/2 but also of other proteins such at PMCA2 that are important for calcium regulation (*Beurg et al., 2010*; *Yamoah et al., 1998*). While we did not observe changes in the distribution of PMCA2 in the stereocilia of mTOMT-deficient hair cells, it might be worthwhile to analyze the extent to which mTOMT regulates PMCA2 activity. Strikingly, reverse polarity currents can be evoked in mTOMT-deficient hair cells by disruption of tip links. Tip-link disruption in mTOMT-deficient hair cells might lead to additional functional perturbation that then affects intracellular calcium levels more severely, thus unmasking reverse-polarity currents. Alternatively, there might be more than one molecular pathway that leads to the activation of reverse polarity currents.

In recent years, remarkable progress has been made in defining the molecular pathogenesis of deafness. One emerging theme is that hair cells and their stereocilia are particularly sensitive to genetic insults (*Dror and Avraham, 2010*; *Richardson et al., 2011*). A significant number of the genes linked to deafness encode proteins that directly affect mechanotransduction (*Kazmierczak and Müller, 2012*; *Zhao and Müller, 2015*). These findings suggest that a subset of hearing disorders can be classified as mechanotransduction diseases. An intriguing open question is whether proteins linked to mechanotransduction in hair cells regulate other mechanosensory

phenomena such as the perception of touch, and whether they play a role in pathological mechanisms such as pain perception.

## Materials and methods

### Mouse strains

Tomt$^{add}$ mice (Tomt$^{m1Btlr}$ RRID:MGI:3805735) were generated using ENU mutagenesis as described (*Du et al., 2008*). CRISPR/Cas9 technology was used to generate mice containing deletions in *Tomt*. Mouse *Tomt* exon 2 sequence was analyzed for potential sgRNA target sites using the Zhang Lab CRISPR design tool (CRISPR.mit.edu). The sgRNA-target genomic DNA sequence (TACGCACCAG-CAGTCGGAAA) was chosen based on proximity to the start codon and a low number of potential off-target sites. Two complementary oligonucleotides encoding the sgRNA-target sequence and its reverse complement were synthesized, annealed and ligated into the *BbsI* restriction enzyme site of pX330 (Addgene #42230). Plasmid DNA was purified from bacterial clones containing sequence-verified px330-*Tomt* sgRNA target sequence. DNA template for in vitro transcription containing T7 promoter, sgRNA target sequence, and tracrRNA was produced by PCR and purified using Phenol-Chloroform precipitation. Chimeric guide RNA was synthesized by in vitro transcription using Ambion MEGAshortscript T7 kit, and RNA was purified using Ambion MEGAclear kit. One-cell embryos were microinjected into pronuclei with 5 ng/µL Cas9 mRNA (TriLink) and 5 ng/µL *Tomt* chimeric guide RNA diluted in nuclease-free TE buffer. After injections, embryos were cultured for 1 day and then implanted into pseudopregnant mice. Pseudopregnant mice were allowed to give birth, offspring resulting from embryo injections were tail-clipped at P21, and genomic DNA was collected. Genomic DNA was screened using PCR and sequencing to determine presence of insertions or deletions. Founder mice containing 12 bp deletions (Tomt$^{D12}$) and 77 bp deletions (Tomt$^{del}$) in exon 2 of *Tomt* were bred with C57BL/6 (RRID:IMSR_JAX:000664) mice, and offspring were screened to verify germ-line transmission of mutations. All animal experiments were approved by the Institutional Animal Care and Use Committee at Johns Hopkins University School of Medicine (#M016M271).

### Auditory brainstem response and distortion product otoacoustic emission testing

Auditory brainstem response (ABR) and distortion product otoacoustic emission (DPOAE) measurements were performed as described in *Schwander et al. (2007)*. Briefly, 3- to 5-week-old mice were anesthetized using Xylazine/Ketamine and transferred into a sound-attenuation chamber while on a heating pad to maintain body temperature. For ABR, an EC1 speaker was placed inside the external ear canal, and electrodes were inserted under the skin at the vertex and pinna while a ground was inserted under the skin near the tail. Click and Pure Tone stimuli were applied starting at an intensity of 90 dB and decreasing stepwise in intensity. Recordings were performed with a TDT System 3 auditory evoked response workstation (Tucker-Davis Technologies, Alachua, FL). Auditory thresholds were analyzed for both ears (for Click and Pure Tone) and for a range of frequencies (for Pure Tone, 8–32 kHz). Using the TDT workstation, the cubic (2f1-f2) distortion product otoacoustic emission (DPOAE) thresholds were determined at a range of frequencies (8–32 kHz), evoked using two equal intensity primary stimuli, applied to the ear canal using two EC1 speakers coupled via an ER10B+ microphone probe (Etymotic Research, Elk Grove Village, IL). Stimuli were increased in intensity from 0 to 90 dB in 5 dB increments to establish thresholds.

### Scanning electron microscopy

Scannin electron microscopy (SEM) methods were performed as described (*Xiong et al., 2012*; *Zhao et al., 2014*, *Zhao et al., 2016*). P7-P8 mice were transcardially perfused with 4% paraformaldehyde (PFA) in SEM buffer containing 0.05 mM HEPES Buffer pH 7.2, 10 mM CaCl$_2$, 5 mM MgCl$_2$, and 0.9% NaCl. After perfusion, temporal bones were removed from the skull and postfixed in 4% PFA in SEM Buffer for 30 min at room temperature with gentle rocking. The temporal bone was placed in SEM buffer and the bony cochlear shell, tectorial membrane and Reissner's membrane were removed, leaving the Organ of Corti on the modiolus. The samples were then placed in SEM buffer containing 4% PFA and 2.5% glutaraldehyde and incubated overnight at RT. The samples

were washed in SEM buffer with no fixative, and then dehydrated in a series of EtOH treatments. Samples were treated to critical point drying using an Autosamdri-815A (Tousimis) or Baltec 030, mounted on stubs with carbon-coated tape, and coated with Iridium or Chromium using a sputter coater (Iridium—sputter coater EMS150TS; Electron Microscopy Sciences, Chromium—Denton Vacuum). Imaging was performed using a Hitachi S-4800-ll Field Emission Scanning Electron Microscope or a Leo/Zeiss Field Emission Scanning Electron Microscope. Inner and Outer Hair cells were selected from mid-apical regions of the cochlear spiral. Tip links were quantified as the numbers of tip links per stereocilial column from Hair Cells.

## Gentamicin treatment

Cochleas from postnatal day 5 (P5) wild-type and mutant animals were dissected in 1x HBSS containing 0.1 mM $CaCl_2$ and 50 µg/mL Ampicillin, sectioned into three equal pieces representing apex, middle and base, and plated on the bottom of 35 mm tissue-culture dishes containing culture media composed of 1x DMEM/F12, 10% Fetal Bovine Serum (FBS) and 50 µg/mL Ampicillin. Cochlear explants were incubated at 37°C with 5% $CO_2$. After 4 hr, the media was replaced with fresh culture media with or without 1 mM Gentamicin. The dishes containing the explants were returned to the incubator for 24 hr, fixed by incubating for 30 min in 1x HBSS containing 1.5 mM $CaCl_2$ and 4% PFA. After fixation, explants were immunostained for MYO7A and phalloidin and imaged to analyze hair cell survival. For catecholamine treatments with Gentamicin, explants were pretreated for 24 hr with catecholamines or inhibitors prior to gentamicin treatment. Media was then replaced with fresh media with or without 1 mM Gentamicin and catecholamines or inhibitors. Explants were treated for an additional 24 hr, followed by fixation, immunostaining and imaging as above.

## Electrophysiology

Mechanotransduction currents were recorded following our published procedures (*Wu et al., 2017*; *Xiong et al., 2012*; *Zhao et al., 2014*). In whole-cell configuration, cochlear hair cells were patched at −70 mV holding potential with a patch clamp amplifier (EPC 10 USB, HEKA Electronics, Lambrecht/Pfalz, Germany). For mechanical stimulation, hair bundles were deflected with a glass probe mounted on a piezoelectric stack actuator (P-885, Physik Instrument, Karlsruhe, Germany). The actuator was driven with voltage steps that were low-pass filtered at a frequency of 10 KHz to diminish the resonance of the piezo stack. The recording chamber was perfused with artificial perilymph (in mM): 144 NaCl, 0.7 $NaH_2PO_4$, 5.8 KCl, 1.3 $CaCl_2$, 0.9 $MgCl_2$, 5.6 glucose, and 10 K-HEPES, pH 7.4. The patch pipette was filled with intracellular solution (140 mM KCl, 1 mM $MgCl_2$, 0.1 mM EGTA, 2 mM Mg-ATP, 0.3 mM Na-GTP and 10 mM K-HEPES, pH 7.2). The membrane potential was measured under current-clamp model by patch-clamping P6 Outer hair cells at 0 pA. The whole-cell outward currents were elicited from P6 Outer hair cells by a series of command potentials from −90 mV to 60 mV for 150 ms in 10 mV steps (5 s intervals) from a holding potential of −70 mV. For nonlinear capacitance, Organs of Corti were dissected from the cochlea of postnatal mice (P16). The apical region was mounted onto a recording dish containing in (mM):100 NaCl, 5.8 KCl, 1.5 $MgCl_2$, 20 TEA, 20 CsCl, 2 $CoCl_2$, 0.001 tetrodotoxin, pH 7.4. The patch pipette was filled with (in mM): 140 CsCl, 5 TEA, 2 MgCl2, 10 EGTA, pH 7.2. Voltage-dependent capacitance was determined by Lock-in function of Patchmaster (HEKA). After compensation of membrane capacitance, Command sinusoid (1.0 kHz, amplitude 10 mV, sampling rate 20 kHz) was summed to voltage ramps ranging from −150 to 100 mV during capacitance measurement, 16 cycles were integrated to generate each data point. The data were fitted with a Boltzmann function: $NLC = C_{lin} + Q_{max}/\{[\alpha * exp((V-V_{1/2})/\alpha)]/[1 + exp((V-V_{1/2})/\alpha)]2\}$, where $C_{lin}$ is the residual linear capacitance, V is the membrane potential, $V_h$ is voltage at half-maximal non-linear charge transfer, $Q_{max}$ is the maximal voltage sensor charge moved through the membrane electrical field, and $\alpha$ is the slope factor of the voltage dependence. Reverse polarity currents were elicited from OHCs using a fluid jet from a pipette (tip diameter of 10–15 µm). Sinusoidal force stimulus was applied to a 27-mm-diameter piezoelectric disc to produce fluid jet. The position of the pipette delivering the fluid jet was positioned at the modiolar side of the hair bundles and adjusted to elicit maximal mechanical stimuli-induced currents. For BAPTA treatments, Organs of Corti were perfused with 5 mM BAPTA for 10 min prior to recording. The sinusoids (40 Hz) was generated with Patchmaster 2.35 software (HEKA) and filtered at 1.0 kHz with 900CT eight-pole Bessel filter (Frequency Devices, Ottawa, IL).

## Whole-mount immunohistochemistry, immunocytochemistry and imaging

Whole mount cochleas were dissected, fixed, and immunostained as described (*Grillet et al., 2009*; *Schwander et al., 2007*; *Senften et al., 2006*). Briefly, temporal bones from postnatal wildtype and mutant mice were dissected away from the skull in 1x HBSS containing 1.5 mM $CaCl_2$, and openings were made in the bony cochlear shell at the apex and through the oval and round windows. The temporal bones were incubated in 1x HBSS containing 4% PFA and 1.5 mM $CaCl_2$ for 30 min at RT. The temporal bones were washed in 1x PBS, the bony cochlear shell was removed, and the Organ of Corti was separated from the modiolus and collected for immunostaining. Reissner's membrane and the Tectorial membrane were removed, and the Organ of Corti was permeabilized in PBS containing 5% BSA and 0.5% Triton for 30 min at RT. After permeabilization, the tissue was blocked in PBS containing 5% BSA for 1 hr. The tissue was incubated in PBS containing 5% BSA and primary antibodies (see below) overnight at 4°C. The tissue was then washed three times in PBS, and incubated for 1–2 hr at RT in PBS containing 5% BSA, secondary antibodies (1:1000, Invitrogen, see below), and phalloidin conjugated to various Alexa Fluor dyes (Invitrogen, phalloidin 488/568/647, 1:500). After secondary antibody incubation, the tissue was washed three times and mounted on slides using ProLong Diamond (Invitrogen). Whole mounts were imaged using 60x and 100x lenses on a widefield fluorescence deconvolution microscope (Deltavision, GE Life Sciences). For immunocytochemistry, Heterologous cells (HEK293 (RRID:CVCL_0045) or COS-7 cells (RRID:CVCL_0224)) were plated to 70–90% confluence in six-well plates. HEK293 cells (CRL-1573, Lot #: 61714301) and COS-7 cells (CRL-1651, Lot #: 59325379) were obtained from ATCC. Cells were authenticated using STR profiling and verified to be free of Mycoplasma contamination. Cells were transfected with plasmids using Lipofectamine 2000, incubated for 24 hr, re-plated on glass coverslips in 24-well plates, and incubated for an additional 24 hr. Cells were fixed in 4% paraformaldehyde in 1X PBS for 10 min, permeabilized in 0.25% Triton for 10 min, and then processed for immunostaining and imaged as above.

Primary antibodies were as follows: Rabbit anti-Myosin VIIa (1:500, Proteus, RRID:AB_10015251), Rabbit anti-Cdh23 (1:500, [*Kazmierczak et al., 2007*; *Siemens et al., 2004*]), Rabbit anti PCDH15-CD2 (1:500, [*Webb et al., 2011*]), Rabbit anti-TMHS/LHFPL5 (1:500, [*Xiong et al., 2012*]), Rabbit anti-PMCA2 (1:200, Abcam, RRID:AB_303878), Rabbit anti-Myc (1:500, Cell Signaling, RRID:AB_490778), mouse anti-Espin (1:200, BD Biosciences, RRID:AB_399174), mouse anti-HA (1:500, Cell Signaling, RRID:AB_331789), mouse anti-Myc (1:8000, Cell Signaling, RRID:AB_331783), Chicken anti-GFP (1:1000, Aves, RRID:AB_10000240). The Rabbit anti-Whirlin antibody (1:500) was generated by Covance (Denver, PA) against a peptide specific for the mouse Whirlin protein. The antibody was affinity-purified against the peptide used as an immunogen.

Secondary antibodies were as follows: Goat anti-Mouse F(ab')2 Alexa Fluor 488 (1:1000, Invitrogen, RRID:AB_2534084), Goat anti-Rabbit F(ab')2 488 (1:1000, Invitrogen, RRID:AB_2534114), Goat anti-Rabbit F(ab')2 Alexa Fluor 555 (1:1000, Invitrogen, RRID:AB_2535851), Goat anti-Rabbit Alexa Fluor 555 Superclonal (1: 1000, Invitrogen, RRID:AB_2536100), Goat anti-Mouse F(ab')2 Alexa Fluor 647 (1:1000, Invitrogen, RRID:AB_2535806).

## Injectoporation

Exogenous DNA plasmids were expressed in cochlear hair cells via injectoporation as published (*Xiong et al., 2012*, *2014*; *Zeng et al., 2016*; *Zhao et al., 2014*). Cochlear explants from P3 wildtype and mutant mice were dissected in media containing 1x HBSS containing 0.1 mM $CaCl_2$ and 50 µg/mL Ampicillin, sectioned into three equal pieces representing apex, middle and base and plated on the bottom of low-walled 35-mm culture dishes containing injectoporation media composed of 1x DMEM/F12 and 50 µg/mL Ampicillin and incubated at 37°C with 5% $CO_2$. After 4–6 hr, dishes containing explants were prepared for injectoporation. Glass electrodes with an inner diameter of 2 µm were used to deliver plasmid (1 µg/uL in 1x HBSS) to the sensory epithelium of explants. Four square wave pulses were applied with an electroporator (ECM 830, Harvard Apparatus) at 1 s intervals with a magnitude of 60 V and duration of 15 ms. After injectoporation, the media in the dishes was changed to fresh culture media containing 1x DMEM/F12, 10% Fetal Bovine Serum (FBS) and 50 µg/mL Ampicillin. Dishes were returned to the 37°C incubator and incubated for 24 hr, followed by fixation, immunostaining, and imaging as above.

### Adeno-associated virus production and In vivo AAV injection

AAV serotype 2/1 viral vectors were produced by PEI transfection in HEK293 cells. *Tomt-HA* or *Tomt-GFP* was cloned into pAAV-CAG-GFP (Addgene #37825) between the XbaI and EcoRI sites. 250 μg pHelper (generous gift of Anton Maximov), 125 μg pAAV-RC1 (Cell Biolabs), and 125 μg pAAV-CAG-*Tomt-HA* or pAAV-CAG-*Tomt-GFP* in OptiMEM were used with 700 μL of 1 mg/mL PEI (Polysciences, Inc) in OptiMEM to transfect ten 150-mm petri dishes containing HEK293 cells at 70% confluency. After 3 days, cells were lysed, collected, and AAV viral particles were purified using AAV-pro kit (Clontech). Purified AAV vectors were stored at −80°C. For in vivo injections, anesthetized P0-P2 mouse pups were injected with a Hamilton syringe with a beveled tip containing 1–2 μL of AAV. The needle was inserted through the skin into the temporal bone, containing the inner ear. After injection, the pups were allowed to recover from anesthesia, returned to their cage and allowed to survive until P7-P8, after which they were sacrificed and cochleas were fixed, immunostained, and imaged as above.

### HPLC quantification of catecholamine levels

Temporal bones from P5 wild-type and $Tomt^{add/add}$ mice were dissected and snap frozen on dry ice in eppendorf tubes. Each tube contained two temporal bones pooled together and represented one animal. For tissue extraction, the frozen temporal bones were homogenized using a tissue dismembrator, in 0.1M TCA, containing 0.1 M sodium acetate, 0.1 mM EDTA, 5 ng/ml isoproterenol (as internal standard) and 10.5% methanol (pH 3.8). Ten microliters of homogenate was set aside for assay of protein concentration using BCA protein assay kit (Thermo Scientific). The samples were then centrifuged and the supernatant was collected for analysis via HPLC.

Catecholamines and other biogenic amine levels were determined by an HPLC assay utilizing an Antec Decade II (oxidation: 0.65) electrochemical detector operated at 33°C. Twenty microliter samples of the supernatant were injected using a Water 2707 autosampler onto a Phenomenex Kintex (2.6u, 100A) C18 HPLC column (100 × 4.60 mm). Biogenic amines were eluted with a mobile phase consisting of 89.5% 0.1M TCA, 0.1 M sodium acetate, 0.1 mM EDTA and 10.5% methanol (pH 3.8). Solvent was delivered at 0.6 ml/min using a Waters 515 HPLC pump. Using this HPLC solvent the biogenic amines elute in the following order: Noradrenaline, Adrenaline, DOPAC, Dopamine, 5-HIAA, HVA, 5-HT, and 3-MT. HPLC control and data acquisition were managed by Empower software. Absolute catecholamine levels were normalized to protein concentration values and reported as ng catecholamine/mg of protein.

The tissue extraction, protein assay and HPLC measurements were performed by the Vanderbilt Neurochemistry Core Facility.

### Co-immunoprecipitation and western blotting

Co-immunoprecipitations were performed as described (*Senften et al., 2006*; *Zhao et al., 2014*, *Zhao et al., 2016*). Briefly, HEK293 cells were transfected with various plasmids using Lipofectamine 2000. After 48 hr, cells were lysed using CoIP buffer containing 50 mM HEPES (pH 7.5), 50–300 mM NaCl, 1.5 mM MgCl$_2$, 10% glycerol, 1% Triton X-100, 1 mM EDTA and a Roche Complete Protease Inhibitor Tablet. After lysis, lysates were centrifuged at 20,000 rcf for 15 min at 4°C. Supernatant was collected and precleared with EZ View Red Protein A Agarose (Sigma). At this point, 10% of the lysate was set aside for use as an input control. The rest of the lysate was immunoprecipitated for 1 hr at 4°C using EZ View Red HA Affinity Gel or EZ View Red Flag M2 Affinity Gel. After immunoprecipitation, the affinity gel was washed three times with CoIP buffer and boiled in Lamelli Buffer containing beta-mercaptoethanol to elute protein complexes. Eluted immunoprecipitated protein was run in parallel with input lysate on 4–20% SDS-PAGE gels (Biorad) and transferred to PVDF membranes using iBlot2 (Thermo Scientific) or using traditional wet transfer. Membranes were blocked for 1 hr with 1% Bovine Serum Albumin (BSA) in 1X TBST (containing 20 mM Tris-HCl pH 7.5, 150 mM NaCl and 0.1% Tween-20). Membranes were incubated with primary antibodies (see below) in 1% BSA in 1X TBST at 4°C overnight. After primary antibody incubation, membranes were washed three times with 1X TBST, followed by incubation for 1 hr in solution containing secondary antibodies (see below) in 1% BSA in 1X TBST. Membranes were washed three times in 1X TBST and then imaged with Clarity or Clarity Max Substrate (Biorad) on a G-Box ECL imager (Syngene).

Primary antibodies were as follows: mouse anti-HA (1:750, Cell Signaling, RRID:AB_331789), mouse anti-Flag M2 (1:500, Sigma, RRID:AB_262044), rabbit anti-MYC (1:500, Cell Signaling, RRID: AB_490778), rabbit anti-HA (1:25000, Abcam, RRID:AB_307019), rabbit anti-PCDH15-CD2 (1:5000, [*Webb et al., 2011*]), and rabbit anti CDH23 (1:5000, [*Siemens et al., 2004*]). Secondary Antibodies: Veriblot (1:5000, Abcam,), Veriblot anti-Mouse (1:5000, Abcam).

## DNA constructs and plasmids

Expression vectors for MYC-TMC1, MYC-TMC2, CDH23, PCDH15-CD2, HA-LHFPL5/TMHS and TMIE-HA were described previously (*Siemens et al., 2004*; *Webb et al., 2011*; *Xiong et al., 2012*; *Zhao et al., 2014*, *Zhao et al., 2016*). pCMV-LRTOMT2-tGFP was obtained from Origene (RG226621). Additional DNA constructs were generated as described below. All constructs were sequence verified.

### pCAGEN-HA-TOMT

Mouse cochlear cDNA was used as a PCR template to amplify *Tomt* coding sequence (based on NCBI Consensus Coding Sequence # CCDS40044.1) using PCR primers containing HA tag, EcoRI, and NotI. *HA-Tomt* was ligated into pCAGEN (Addgene #11160) digested with EcoRI and NotI. PCR primers were: (i) 5'-AAAAAAAAAAGAATTCGCCACCATGTACCCATACGATGTTCCAGATTACGC TTCCCCTGCCATTGCACT-3' containing EcoRI sequence, Kozak sequence and HA tag; (ii) 5'-AAAAAAAAAAGCGGCCGCTCAGCCGGGTCCAGTATAGG-3' containing NotI sequence.

### pCAGEN-TOMT-GFP

Mouse cochlear cDNA was used as a PCR template to amplify *Tomt*, without the final stop codon (*TomtΔstop*). pPBCAG-GFP (generous gift of Joseph LoTurco) was used as a PCR template to amplify *GFP* (*GFPΔstart*), without the start codon. Assembly PCR was used with *TomtΔstop* and *GFPΔstart* as templates to create an in-frame fusion, and the product of this reaction was ligated into pCAGEN digested with EcoRI and NotI. PCR Primers were: (i) To amplify *TomtΔstop*: 5'-AAAAAAAAAAGA ATTCGCCACCATGTCCCCTGCCATTGCAC-3' containing EcoRI sequence, and 5'-AACAGCTCCTCGCCCTTGCTCACGCCGGGTCCAGTATAGGTGA'−3; (ii) To amplify *GFPΔstart*: 5'-GTGAGCAAGGGCGAGGAG-3, and 5'-AAAAAAAAAAGCGGCCGCTTA CTTGTACAGCTCG TCCATGCC-3', containing NotI sequence; (iii) Assembly primers to amplify *Tomt-GFP* 5'-AAAAAAAAAAGAATTCGCCACCATGTCCCCTGCCATTGCAC-3' containing EcoRI sequence 5'-AAAAAAAAAAGCGGCCGCTTACTTGTACAGCTCGTCC ATGCC-3', containing NotI sequence.

### pAAV-CAG-TOMT-GFP

pCAGEN-TOMT-GFP was used as a PCR template to amplify *Tomt-GFP*, using primers containing XbaI and EcoRI. Tomt-GFP was ligated into pAAV-CAG-GFP (Addgene #37825) digested with XbaI and EcoRI and gel purified to remove GFP. PCR Primers were: 5'-AAAAAAAAAATCTAGAGC-CACCATGTCCCCTGCCATTG CAC-3' containing an XbaI sequence. 5'-AAAAAAAAAAGAATTC TTACTTGTACAGCTC GTCCATGCC-3' containing an EcoRI sequence.

### pAAV-CAG-TOMT-HA

Mouse cochlear cDNA was used as a PCR template to amplify *Tomt*, using PCR primers containing HA tag, XbaI and EcoRI. *Tomt-HA* was ligated into pAAV-CAG-GFP (Addgene #37825) digested with XbaI and EcoRI and gel purified to remove GFP. PCR primers were: (i) 5'-AAAAAAAAAATC TAGAGCCACCATGTCCCCTG CCATTGCAC-3' containing XbaI sequence and Kozak sequence; and (ii) 5'-AAAAAAAAAAGAATTCTCAAGCGTAATCTGGAACATCGTATGGGTAGCCGGGTCCAGTA TAGGT-3' containing EcoRI sequence and HA-tag.

### pAAV-CAG-COMT-GFP

Mouse cochlear cDNA was used as a PCR template to amplify *Comt*, without the final stop codon (*ComtΔstop*). pPBCAG-GFP (generous gift of Joseph LoTurco) was used as a PCR template to amplify *GFP* (*GFPΔstart*), without the start codon. Assembly PCR was used with *ComtΔstop* and *GFPΔstart* as templates to create an in-frame fusion, and the product of this reaction was ligated

into pAAV-CAG-GFP (Addgene #37825) digested with XbaI and EcoRI and gel purified to remove GFP.

PCR Primers were: (i) To amplify *ComtΔstop*: 5'-AAAAAAAAAATCTAGAGCCACCAT GCTG TTGGCTGCTGTCTCAT-3' containing XbaI sequence, and 5'-CTCCTCGCCCTT GCTCACGGAC TTCACGGGGCTGCT-'3; (ii) To amplify *GFPΔstart*: 5'-AGCAGCCCC GTGAAGTCCGTGAG-CAAGGGCGAGGAG-3', and 5'-AAAAAAAAAAGAATTCTTACTT GTACAGCTCGTCCATGCC-3', containing EcoRI sequence; (iii) Assembly primers to amplify *Comt-GFP*: 5'-AAAAAAAAAATC TAGAGCCACCATGCTGTTGGCTGCTGTCT CAT-3' containing XbaI sequence, and 5'-AAAAAAAAAAGAATTCTTACTTGTACAGCT CGTCCATGCC-3', containing EcoRI sequence.

### pCAGEN-TOMT-Y108A-GFP

Site-directed mutagenesis to modify pCAGEN-Tomt-GFP to pCAGEN-Tomt-Y108A-GFP was performed according to instructions using QuikChange II XL Site-Directed Mutagenesis Kit (Agilent). Mutations in *Tomt-GFP* DNA sequence were t322g and a323c. PCR primers to make t322g and a323c mutations:

(i)5'-GTGTAGAGTATCCACAGGCTGTGCCCAGCTCCAGCA-3' and (ii) 5'-TGCTGGAGCTG GGCACAGCCTGTGGATACTCTACAC-3'.

### pAAV-CAG-TOMT-MT-GFP

pCAGEN was modified to contain a multiple cloning site containing NheI, SacI, XmaI, and KpnI sites followed by GFP (from pPBCAG-GFP) with a deleted start codon (*GFPΔstart*) between EcoRI and NotI sites. pCAGEN-TOMT-GFP was used as a template to amplify the putative *Tomt* methyltransferase (MT) domain using PCR primers containing EcoRI and KpnI. *Tomt* MT domain was ligated into modified pCAGEN-MCS-GFPΔstart was digested with EcoRI and KpnI. PCR primers were: 5'-AAAAAAAAGAATTCGCCACCATGGTGCTGGAGCTGGGCA-3', containing an EcoRI sequence, and 5'-AAAAAAAAGGTACCCAATACAGTGGCGCCATGT- 3', containing a KpnI sequence.

## Data analysis

Data analysis was performed using Excel (Microsoft) and Igor pro 6 (WaveMetrics, Lake Oswego, OR). Calcium signal ($\triangle$F/F) was calculated with the equation: $(F-F_0)/F_0$, where $F_0$ is the averaged fluorescence baseline at the beginning. Transduction current-displacement curves (I(X)) were fitted with a three-state Boltzmann model (*Grillet et al., 2009*). Student's two-tailed unpaired t test was used to determine statistical significance (*$p<0.05$, **$p<0.01$, ***$p<0.001$).

## Acknowledgements

We thank members of the laboratory for comments and criticisms. We are grateful to Cynthia Ramos, Kaitlyn Nguyen, Guihong Peng and Michelle Monroe for assistance with mouse genotyping. We thank Malcolm Wood and Theresa Fassel at the TSRI Microscopy Core; and Michael Delannoy and Barbara Smith at the Johns Hopkins Microscopy Core for assistance with SEM. We thank the Vanderbilt Neurochemistry Core Facility for assistance with HPLC. This work was supported by the NIH (UM; DC005965, 124789, 124906; CLC; DC015724; AL; DC005211, DC00023) and the David M Rubenstein Fund for Hearing Research. UM is a Bloomberg Distinguished Professor for Neuroscience and Biology.

## Additional information

### Competing interests

UM: Ulrich Mueller is a founder of Decibel Therapeutics. The other authors declare that no competing interests exist.

### Funding

| Funder | Grant reference number | Author |
| --- | --- | --- |
| National Institute on Deafness and Other Communication | DC015724 | Christopher L Cunningham |

| | | |
|---|---|---|
| Disorders | | |
| National Institutes of Health | DC005211 | Amanda Lauer |
| National Institutes of Health | DC000023 | Amanda Lauer |
| National Institute on Deafness and Other Communication Disorders | DC005965 | Ulrich Müller |
| National Institute on Deafness and Other Communication Disorders | 124789 | Ulrich Müller |
| Foundation for the National Institutes of Health | 124906 | Ulrich Müller |
| Rubenstein fund | | Amanda Lauer |

The funders had no role in study design, data collection and interpretation, or the decision to submit the work for publication.

## Author contributions

CLC, Conceptualization, Data curation, Formal analysis, Funding acquisition, Investigation, Methodology, Writing—original draft, Writing—review and editing; ZW, Conceptualization, Data curation, Formal analysis, Investigation, Methodology, Writing—review and editing; AJ, Conceptualization, Data curation, Formal analysis, Investigation, Methodology; BZ, Conceptualization, Data curation, Formal analysis, Funding acquisition, Investigation, Methodology; KS, Formal analysis, Investigation, Methodology; SH-P, Investigation, Methodology; AL, Formal analysis, Supervision, Investigation; UM, Conceptualization, Resources, Formal analysis, Supervision, Funding acquisition, Investigation, Writing—original draft, Project administration, Writing—review and editing

## Author ORCIDs

Christopher L Cunningham, http://orcid.org/0000-0002-9619-4190
Amanda Lauer, http://orcid.org/0000-0003-4184-7374
Ulrich Müller, http://orcid.org/0000-0003-2736-6494

## Ethics

Animal experimentation: All animal experiments were approved by the Institutional Animal Care and Use Committee at Johns Hopkins University School of Medicine (#M016M271).

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
