## [Decision Letter]

Thank you for submitting your article "The Catecholamine Methyltransferase mTOMT is Critical for Mechanotransduction by Cochlear Hair Cells" for consideration by *eLife*. Your article has been reviewed by three peer reviewers, and the evaluation has been overseen by a Reviewing Editor and Richard Aldrich as the Senior Editor. The following individuals involved in review of your submission have agreed to reveal their identity: David Raible (Reviewer #1); Gregory I Frolenkov (Reviewer #2).

The reviewers have discussed the reviews with one another and the Reviewing Editor has drafted this decision to help you prepare a revised submission.

Cunningham et al. perform comprehensive analysis of roles of the catecholamine methytransferase family gene mTOMT in mechanosensory hair cells. Using a combination of in vivo and in vitro analyses, histology and physiology, they demonstrate a requirement for mTOMT in hair cell mechanotransduction. The authors show that mTOMT mouse mutants have profound hearing loss through ABR recordings and that their outer hair cells fail to function shown through a loss of DPOAEs. These defects appear to be due to a loss of mechanotransduction in Tomt mutant mice. It has previously been hypothesized that TOMT's role in hair cells was due to impaired catecholamine degradation. The authors provide data that this is likely not the case due to the fact that application of exogenous catecholamines and pharmacological inhibition of catecholamine degradation have no effect on hair cell activity. Also, a TOMT construct with a point mutation eliminating TOMTs methyltransferase activity is able to rescue TOMT mutants to a similar extent as wildtype TOMT. They additionally show that TOMT is able to interact with a number of proteins important for mechanotransduction activity including PCDH15, LHPL5, TMIE and TMC2 and that TMC1 and TMC2 fail to properly localize to the stereocilia in *Tomt* mutants.

Even though the mechanism of this regulation is unclear, this is the first identified molecule that may regulate trafficking of the MET channel components in the hair cells. Therefore, the study is significant despite all drawbacks described below.

Essential revisions:

While this mislocalization is presumably responsible for mechanotransduction defects seen in *Tomt* mutants it is not sufficient for the emergence of reverse polarity currents that are normally seen in Tmc mutants. This major take-home message is based on three panels (Figure 8), out of which only the panels I and J show somewhat convincing TMC2 labeling in one stereocilia bundle of an outer hair cell (OHC) in each panel. This is not very compelling for a crucial result of the study. Some sort of statistical analysis is needed. If such analysis is not possible, then perhaps presenting the images of all transfected cells from these experiments in the supplement could make the authors' claim more convincing.

Where is mTOMT localized within the cell? Does it promote overall stability or production of TMC1/2? That is, are overall levels of TMC1/2 higher in cells with mTOMT activity or is there differential localization?

It is surprising that characterization of the OHC function was limited to mechanotransduction, even though TOMT deficiency may affect trafficking of other proteins besides TMC1 and TMC2. In fact, the accompanying manuscript from Teresa Nicolson's group has indications for that. Similarly, Ahmed et al. suggested the role LRTOMT at the basolateral plasma membrane of OHCs (Nat. Genetics, 2008). The "low hanging fruits" are the resting membrane potential, voltage-gated ion currents, and electromotility (non-linear capacitance) of the OHCs. Without knowing the physiological condition of OHCs, it is hard to determine whether the lack of proper trafficking of TMC1 and TMC2 is a primary or secondary effect of TOMT deficiency.

The MET function was investigated in OHCs (Figure 5) while tip link count and localization of tip-link proteins (Figure 6) were performed in the inner hair cells (IHCs). Although the reviewers doubt that TOMT deficiency affects differently tip links in the inner and outer hair cells, this inconsistency negates the authors' conclusion on the loss of MET current in TOMT-deficient hair cells without noticeable effects on the tip links.

Figure 6. The most essential data on these panels is the localization of tip link proteins, CDH23 and PCDH15-CD2. Yet, localization of CDH23 was investigated in add/add mice while localization of PCDH15-CD2 was explored in del/del mice. Perhaps, they are similar but that needs to be shown. To make an argument about the loss of MET without changes in tip links, the PCDH15-CD2 localization has to be explored in add/add mice, since MET recordings were performed in this mouse strain (Figure 5).

Figure 8: The hair cells may survive gentamicin treatment only if the resting MET current is completely abolished. In case of partial inhibition of MET current, one could still expect degeneration of hair cells after 24-hour exposure to gentamicin. Therefore, these data are not very informative as they indicate only that the MET current was not inhibited completely. It would be more helpful to provide the actual records of the MET current. As mentioned in the manuscript, the MET current recordings are already available.

---

## [Author Response]

*Essential revisions:*

*While this mislocalization is presumably responsible for mechanotransduction defects seen in Tomt mutants it is not sufficient for the emergence of reverse polarity currents that are normally seen in Tmc mutants. This major take-home message is based on three panels (Figure 8), out of which only the panels I and J show somewhat convincing TMC2 labeling in one stereocilia bundle of an outer hair cell (OHC) in each panel. This is not very compelling for a crucial result of the study. Some sort of statistical analysis is needed. If such analysis is not possible, then perhaps presenting the images of all transfected cells from these experiments in the supplement could make the authors' claim more convincing.*

We provide now in Figure 8 a quantification of the data showing robust localization of MYC-TMC1 and MYC-TMC2 to the hair bundles in Tomt^del/+^ mice but not in Tomt^del/del^ mice. We would like to emphasize that consistent with findings by other laboratories, it is much more difficult to express TMC1 in hair cells compared to TMC2 (for example see the original publication on TMC1 in hair cells by Kawashami et al., 2011). Thus, the overall levels of TMC1 in the hair bundles are lower compared to TMC2, but TMC1 can clearly be localized to the bundle. We have revised the text to indicate that expression levels for TMC1 are low compared to TMC2.

*Where is mTOMT localized within the cell? Does it promote overall stability or production of TMC1/2? That is, are overall levels of TMC1/2 higher in cells with mTOMT activity or is there differential localization?*

We have carried out additional experiments and have revised the text and figures accordingly: “We have now carried out a more detailed analysis of mTOMT localization in hair cells as well as in heterologous cells. […] On rare occasions (<1/10 cells) we observed expression of mHA-TOMT in stereocilia (data not shown). However, the significance of this finding is currently unclear.”

We also would like mention that we have used CRISPR technology to generate a mouse line where the endogenous mTOMT protein was tagged with HA. The advantage of the mouse line is over injectoporation is that gene expression is driven by the genomic *mTomt* locus thus providing a greater chance that the transgene is expressed at normal levels. Consistent with the data shown in the manuscript, we observed mTOMT expression throughout the cell body of hair cells. We could show the data in the current manuscript but would prefer not to do so because we generated the mice as the foundation for a new study to further define the mechanisms of TOMT function in hair cells. Nevertheless, we provide here for inspection a figure to the reviewers that supports the conclusions put forward in the current manuscript.

The reviewer also wonders whether mTOMT might regulate endogenous levels of TMC1/2. While we cannot exclude this possibility, it does not affect the conclusion that TMC1/2 does not travel to stereocilia in *mTomt* mutant mice since we used overexpression of epitope tagged proteins to support our conclusion (Figure 8). However, we have revised the text to indicate that we cannot exclude that mTOMT might also regulate TMC1/2 levels. “Finally, while our data show that TMC1 and TMC2 when overexpressed do not travel effectively into the hair bundles of TOMT-deficient hair cells, TOMT might affect endogenous TMC1/2 also in other ways, for example by affecting protein levels of the two transmembrane proteins.”

*It is surprising that characterization of the OHC function was limited to mechanotransduction, even though TOMT deficiency may affect trafficking of other proteins besides TMC1 and TMC2. In fact, the accompanying manuscript from Teresa Nicolson's group has indications for that. Similarly, Ahmed et al. suggested the role LRTOMT at the basolateral plasma membrane of OHCs (Nat. Genetics, 2008). The "low hanging fruits" are the resting membrane potential, voltage-gated ion currents, and electromotility (non-linear capacitance) of the OHCs. Without knowing the physiological condition of OHCs, it is hard to determine whether the lack of proper trafficking of TMC1 and TMC2 is a primary or secondary effect of TOMT deficiency.*

We agree with the assessment of the reviewer and have included additional experiments to address these points: (i) In Figure 5—figure supplement 1 we show data for resting membrane potential, voltage gated ion currents and non-linear capacitance. There was no difference between controls and *Tomt^add/add^* mutants; (ii) in Figure 6 we show that the distribution of CDH23, PCDH15-CD2, PMCA2, MYO7A, espin and whirlin is not altered in the mutant hair cells; (iii) in Figure 8 we show that the distribution of LHFPL5 and TMIE is not affected in the mutant hair cells.

Naturally, we cannot exclude that other proteins not studied here might be affected by mTOMT-deficiency, but there is an incredible specificity for effects on TMC1/2 without affecting other components of the mechanotransduction machinery of hair cells (CDH23, PCDH15-DC2, MYO7A, LHFPL5, TMIE) and key proteins of hair bundles (PMCA2, espin, whirlin).

*The MET function was investigated in OHCs (Figure 5) while tip link count and localization of tip-link proteins (Figure 6) were performed in the inner hair cells (IHCs). Although the reviewers doubt that TOMT deficiency affects differently tip links in the inner and outer hair cells, this inconsistency negates the authors' conclusion on the loss of MET current in TOMT-deficient hair cells without noticeable effects on the tip links.*

In the revised version we confirm quantitatively that tip-link numbers are not affected in OHCs of *Tomt* mutants (Figure 6). We also show that tip-link components CDH23 and PCDH15-CD2 localizations are unaffected in OHCs of *Tomt* mutants. (Figure 6—figure supplement 1).

*Figure 6. The most essential data on these panels is the localization of tip link proteins, CDH23 and PCDH15-CD2. Yet, localization of CDH23 was investigated in add/add mice while localization of PCDH15-CD2 was explored in del/del mice. Perhaps, they are similar but that needs to be shown. To make an argument about the loss of MET without changes in tip links, the PCDH15-CD2 localization has to be explored in add/add mice, since MET recordings were performed in this mouse strain (Figure 5).*

Revised Figure 6 now contains the localization data from add/add mice.

*Figure 8: The hair cells may survive gentamicin treatment only if the resting MET current is completely abolished. In case of partial inhibition of MET current, one could still expect degeneration of hair cells after 24-hour exposure to gentamicin. Therefore, these data are not very informative as they indicate only that the MET current was not inhibited completely. It would be more helpful to provide the actual records of the MET current. As mentioned in the manuscript, the MET current recordings are already available.*

The electrophysiological data are now provided in Figure 7—figure supplement 1.